# The Influence of SGD noise on Lottery Ticket Performance

## Abstract

The Lottery Ticket Hypothesis states that there exist sparse subnetworks (called 'winning' Lottery Tickets) within dense networks that, when trained under the same regime, achieve similar or better validation accuracy as the dense network. It has been shown that for larger networks and more complex datasets, an additional pretraining step is required for winning lottery tickets to be successfully found. Previous work linked the amount of pretraining required to a measurement of instability to SGD noise. In this paper, we take a closer look at the training hyperparameters that influence SGD instability during normal training and link this to the ability to find 'winning' tickets. We show that this pretraining step is not always necessary to find winning tickets, but that SGD instability can be reduced by smart hyperparameter selection. While tickets found with less unstable hyperparameters at low sparsity achieve less validation accuracy than more unstable approaches, at higher sparsities they suffer less accuracy degradation and consequently outperform those found with more unstable hyperparameters. We furthermore discover that tickets found with less instability to SGD noise have as unexpected side effect that they encode powerful features for classification in the untrained weights. We show that these features do not emerge when extracted under more unstable hyperparameter settings, and that they are transferable to different datasets, as well as enable faster training of the resulting tickets.

## 1 Introduction

In a recent study by Sevilla et al. (2022) it was calculated that the computational resources to train a state-of-the-art neural network model have doubled roughly every 4 to 9 months. Since then, Large Language Models have become more ubiquitous, leading to the reality that these models can only be trained by the largest research centra and corporations, as they require enormous amounts of computation to train.

There exist however techniques to reduce the cost of training a model. These focus either on intelligently selecting the training data for the model with techniques such as coreset selection (Guo et al., 2022) or dataset distillation (Wang et al., 2018; Zhao et al., 2021), or reducing the number of trainable parameters. This reduction can be achieved in a number of ways, such as using a larger model to guide a smaller model via Knowledge Distillation (Hinton et al., 2015), training multiple small models in parallel (Zhang et al., 2018), weight quantization (Vanhoucke et al., 2011), or generating sparse networks (Han et al., 2015; Li et al., 2017; Frankle & Carbin, 2019; Lee et al., 2019; Ramanujan et al., 2020).

A promising approach to generate sparse networks is Lottery Ticket Hypothesis (LTH) (Frankle & Carbin, 2019). This hypothesis states that within a randomly initialized dense network, there exist a sparse network that can be trained to similar accuracy as the dense network under the same training regime. While the commonly-used approach to find these networks (called 'Winning Tickets') is an intensive iterative procedure called Iterative Magnitude Pruning (IMP), the sparsity of the resulting networks cannot be matched by other approaches without compromising the final accuracy. As such, techniques that allow for speeding up IMP, or increasing our understanding of the functionality behind it are highly valuable.

It was noted in Frankle et al. (2020) that Lottery Tickets for more complex dataset-network settings could not be found directly in random initializations. Rather, they discovered that in such cases, the IMP needed to be applied on a lightly pretrained model, which they term late rewinding, or Lottery Ticket Rewinding

(LTR). The authors link this phenomenon to instability to SGD noise, and quantify this instability by measuring the maximum error on the interpolation path between two solutions found via SGD starting from the same initialization. The application of this pretraining step has been treated as a given, with limited research on other approaches to limit SGD noise and how it impacts the existence of winning lottery ticket. In this study we explore whether it is possible to find winning tickets in complex settings without applying this pretraining step, thus returning to the original definition of the Lottery Ticket Hypothesis. For this we aim to limit the accumulation of SGD noise throughout the pruning process, by modifying hyperparameters associated with instability.

### 1.1 Contributions

- By calculating the instability to SGD noise of a dense network for different hyperparameters settings, we show that certain hyperparameters can limit SGD instability, but that this is linked to a reduced generalization error.

- By considering these hyperparameters in IMP, we show that we can extract 'winning' tickets without pretraining. Additionally, when going sparser, these tickets suffer from less accuracy degradation, opposed to other hyperparameter configurations.

- As an additional bonus, we show that these Lottery Tickets found under more stable hyperparameter configurations can outperform tickets found with late rewinding under certain unstable hyperparameter configurations, albeit at extreme sparsities.

- We discover that tickets found with increased SGD stability can function as feature extractors when frozen, showing that IMP is a form of feature selection. These features additionally show a remarkable capacity for generalization to other datasets.

- Building upon the previous observation, we notice that these tickets can be trained to a surprisingly competitive accuracy with only a small subset of the dataset.

## 2 Related Work

### 2.1 The Lottery Ticket Hypothesis

**Analysis and extensions.** Since the introduction of the Lottery Ticket Hypothesis (LTH), and the Iterative Magnitude Pruning (IMP) algorithm by Frankle & Carbin (2019), the phases that make up IMP have been analyzed and extended. Starting with Zhou et al. (2019) which listed three distinct components in the *Mask Search* phase, namely the Mask Criterion, the Mask-1 action, and the Mask-0 action, and performed an ablation study across these components. Frankle et al. (2020) discovered that Lottery Tickets were unable to be found in more complex settings. This was remedied by introducing late rewinding, applying IMP on a lightly pretrained network. Both Zullich et al. (2021); You et al. (2020) introduce methodologies to speed the Mask Search phase. In the case of Zullich et al. (2021), this is with the introduction of Accelerated Iterative Magnitude Pruning (AIMP), while You et al. (2020) introduces Early-bird tickets. Paul et al. (2022) introduces a loss landscape geometry interpretation of IMP and used this to answer several questions related to the working of IMP.

Our analysis is also centered on the *Mask Search* phase, and more specifically the parameters which influence the instability of this phase. This can be seen as a continuation of an experiment from Maene et al. (2021), which shows that for a sufficiently large batch size, winning tickets can be still be found where previously late rewinding was required.

**Lottery Tickets Features.** A number of studies has been conducted on which inductive features are contained in Lottery Tickets. Studies such as Morcos et al. (2019); Mehta (2019); Desai et al. (2021); Chen et al. (2021) train a ticket extracted for one dataset or data domain from scratch and tested it on another target dataset or data domain. They all show that these trained lottery tickets can generalize as well as a trained dense model on the target dataset.

In this paper, we go a step further and measure the expressivity of the untrained ticket by finetuning a linear layer directly on the target dataset.

## 2.2 (Linear) Mode Connectivity.

Garipov et al. (2018); Draxler et al. (2018) simultaneously discovered that neural network solutions can be connected in the loss landscape via a continuous path such that the error across the path does not exceed a certain threshold. In Garipov et al. (2018), this has been shown to be valid for polynomial paths with a single bend and quadratic Bezier curves. Linear Mode Connectivity goes a step further and requires the path to be linear. Generally, this property has been observed in two distinct cases. The first involves two 'child' models that continue training with different SGD noise from a slightly trained initialization. This paradigm has been introduced in Frankle et al. (2020). Fort et al. (2020) uses the Neural Tangent Kernel to show that early in training the chaotic behavior of SGD determines a loss basin, in which later iterations optimize, thus providing empirical evidence for the emergence of linear mode connectivity. The second approach introduced in Entezari et al. (2022) posits that with intelligent weight permutation, all solutions found by SGD lie in the same loss basin, i.e., are linear connected without error barrier. Ferbach et al. (2024) uses optimal transport to prove this with some added restrictions on the weight distribution.

In this research, we study the impact of different hyperparameters on Linear Mode Connectivity, by quantifying instability as defined in Frankle et al. (2020).

## 3 Methodology

We start from commonly used training configurations (see Appendix G in the appendix) for the Lottery Ticket Hypothesis, taken from Frankle et al. (2021), and study the impact of different hyperparameters on train-time metrics, such as validation accuracy, Linear Mode Connectivity (Frankle et al., 2020), forgetting scores (Toneva et al., 2019), and convergence speed. In the following paragraphs we will first highlight the specific hyperparameters we study, and next we list the train-time metrics we measure.

## 3.1 The Lottery Ticket Hypothesis

---

**Algorithm 1** Iterative Magnitude Pruning with late rewinding.

1: Initialize a neural network with weights $\theta_0 \in \mathbb{R}^d$.
2: Initialize pruning mask $M = 1^d$.
3: Train $\theta_0$ for $p$ steps to $\theta_p$.  ▷ *Pretraining*
4: **for** $n \in \{1, \ldots, N\}$ **do**  ▷ *Mask Search*
5:   Train $M \odot \theta_p$ to convergence.
6:   Prune the k% unpruned weights with lowest magnitude.
    Let $M[i]$=0 if the corresponding weight $i$ is pruned.
7: **end for**
8: Train the final network $M \odot \theta_p$.  ▷ *Sparse Training*

---

Defined by an iterative approach (see Algorithm 1), the procedure to generate Lottery Tickets consists of two (or three) phases. In the optional *Pretraining* phase, the network with weights $\theta_0$ is lightly trained to provide a more stable rewind point (Frankle et al., 2020) for the next phase, resulting in weights $\theta_p$. The second phase is a *Mask Search* phase, in which the sparse network is trained to convergence with the same training configuration as the dense network. The goal of this training is to identify parameters for pruning by determining the lowest magnitude parameters at the end of this phase. Next, a fixed percentage (k) of the lowest magnitude parameters are pruned by setting the pruning mask $M$ for those weights to 0, and the remaining weights are reset to the rewind point (either the initialization $\theta_0$ or the pretrained weights $\theta_p$). These steps are repeated N times. Finally, in the *Sparse Training* phase the ticket is trained until completion, after which the network is usable in prediction. A winning lottery ticket is then defined as a pruned network that can attain similar (or better) validation accuracy as the dense network in commensurate training.

It has been shown in Frankle et al. (2021); Vischer et al. (2022) that the trainability of a ticket is dependent on both the initialization and the specific pruning mask. This directly translates to a dependency on the *Pretraining* phase for the initial weights and a dependency on the *Mask Search* phase for the pruning mask. In this research we specifically focus on the hyperparameters used in the *Mask Search* phase. As a comprehensive grid search on all possible parameters is prohibitively expensive due to the costly, iterative nature of this procedure, we instead focus our attention on a few parameters and their impact. This will allow us to better understand the impact on the trainability and predictive quality of the resulting lottery tickets.

**Batch Size.** In the case of large datasets, it is often infeasible to use the whole dataset during each update step of the network. As such, it is common to use minibatches of the dataset to update a network iteratively via Stochastic Gradient Descent. Intuitively, if the batch size is smaller, then the resulting gradient is more influenced by the subsampling of the dataset, as the batch composition can be dramatically different from the composition of the full dataset. However, it has been shown in Keskar et al. (2017) that smaller batch sizes positively impact the generalization of a neural network on unseen data.

**Momentum.** Introduced as a technique to speed up gradient descent, momentum allows gradient information to be carried over from previous batches in the weight update by employing a factor $\mu \in [0,1]$ to weight the gradient information of the previous batch. As such, the updates function similar to a exponential moving average (see Equations (1) and (2)). The application of momentum has been demonstrated to lead to better generalizing networks, and it has been posited by Jelassi & Li (2022) that this is due to the resulting classifiers generalizing on small-margin samples, rather than memorizing those samples. The authors additionally argue that the impact of momentum is more significant with higher batch sizes.

$$v_{t+1} = \mu v_t - \epsilon \nabla f(\theta_t) \tag{1}$$
$$\theta_{t+1} = \theta_t + v_{t+1} \tag{2}$$

**Training duration.** Training a network is a process with many sources of noise such as mini batch gradients, data augmentation and more. It is evident that this noise accumulates during repeated training epochs. As such, one method to limit SGD noise during training of a network is to simply limit the computational budget of the training phase, rather than tackling the sources of noise.

## 3.2 Train-time Metrics

**Instability to SGD noise.** Introduced in Frankle et al. (2020), this metric measures the relative increase in classification error on a linear interpolating path between two sets of network weights trained from the same initialization with different sets of SGD noise. An initialization is said to be unstable if there is a significant error barrier along the interpolation path. Given two networks $W_1$, $W_2$, with the validation error of a network measured as $\mathcal{E}(W)$, then the error barrier across an interpolating path between $W_1$ and $W_2$ for $\alpha \in [0,1]$ is defined in Equation (3).

$$sup_\alpha \; \mathcal{E}(\alpha W_1 + (1-\alpha)W_2) - \frac{\mathcal{E}(W_1) + \mathcal{E}(W_2)}{2} \tag{3}$$

In practice, as defined by Frankle et al. (2020), a significant error barrier is seen as being higher than 2%. In this case, the network is seen as *unstable* to SGD noise, otherwise it is seen as *stable* to SGD noise.

**Forgetting events.** During training, a network learns to predict labels for its input samples. However, the learning process is not monotonic, meaning that if a sample is predicted correctly at some iteration $i$, it is possible that at iteration $i + k$ it is no longer correctly predicted. To measure these 'forgetting events', Toneva et al. (2019) introduced forgetting scores, which record whether a sample is learned correctly each time the network is fed the sample. Samples that are never forgotten once learned are called 'unforgettable samples'.

**Convergence speed.** By calculating the area under the error curve during a training run of $T$ epochs, we can measure the convergence speed of a training configuration. Specifically, we calculate the error $\mathcal{E}(W^0)$

before training and after every epoch ($\mathcal{E}(W^{ep})$). We then average $\mathcal{E}(W^{ep}) - \mathcal{E}(W^0)$ over the duration of the training. To account for different optimal accuracies, we rescale this value by dividing with $\mathcal{E}(W^T) - \mathcal{E}(W^0)$.

$$\frac{1}{T} \sum_{ep=1}^{T} \frac{\mathcal{E}(W^0) - \mathcal{E}(W^{ep})}{\mathcal{E}(W^0) - \mathcal{E}(W^T)} \tag{4}$$

### 3.3 Experimental setup

For the experiments, we use ResNet-18 (He et al., 2016) trained on CIFAR-10 (Krizhevsky et al., 2009), CIFAR-100 (Krizhevsky et al., 2009), VGG16 (Simonyan & Zisserman, 2014) trained on CIFAR-100, and ResNet-34 (He et al., 2016) trained on TinyImageNet (Le & Yang, 2015). We resort to these datasets and networks, as they are commonly used within the literature. For the transferability experiments in Section 5.1, we additionally use MNIST (LeCun et al., 2010) and EuroSAT (Helber et al., 2019). Most experiments in the main paper list the results for ResNet-34 + TinyImageNet unless otherwise indicated, as this is the most complex setting, but we list results for the other settings in the appendix. Unless otherwise mentioned, each experiment is repeated for three runs.

We use configurations from T et al. (2022) as a starting point. More specifically, this means that we use a total training budget of 200 epochs in each iteration, after which we prune 20% of the weights. In the case of late Rewinding, this budget includes a *Pretraining* phase of 2 epochs. Training is done by minimizing the Cross-Entropy Loss with SGD, starting with a learning rate of 0.1 (or 0.2 for TinyImageNet) which is cosine annealed, and a weight decay of 5E-4. A full breakdown of the training configuration can be found in Appendix G of the appendix.

## 4 Experiments

### 4.1 Instability in Dense Networks

We explore the influence of the momentum and batch size parameters on train-time performance of ResNet-34 models on TinyImageNet. For this, we start with a randomly initialized network and use different sets of SGD noise to achieve three versions of a trained network. In Figure 1 we show the average pairwise interpolation curves and error barriers for the three possible combinations. These results show that when starting from the random initialization (without late rewinding), no configuration can be deemed stable to SGD noise. When starting from a set of pretrained weights, the error barrier decreases significantly for the approaches without momentum ($\mu = 0.0$, solid lines), but only marginally decreases for those with momentum ($\mu = 0.9$, dotted lines). Hyperparameter configurations that lead to less instability also lead to a (slightly) reduced generalization (see 0.00% sparsity column in Table 2). This hints at a relation between instability during the training of dense neural networks and the final performance. In Appendix A of the appendix we further explore the amount of pretraining required for each set of hyperparameters to achieve stability, where we show that settings with lower batch size, or momentum require more pretraining to be stable.

**Example Forgetting.** When observing the results of the example forgetting metric (Table 1), we can see a link between an increase in forgetting events on one side and a decrease in unforgettable samples, and an increase in SGD instability on the other side. In general, the application of momentum seems to have a much higher impact on the number of forgetting events than a different batch size has. We hypothesize that the frequent forgetting and re-learning of certain samples might steer the model to better generalization. As observed in Jelassi & Li (2022), momentum can lead models to generalize rather than memorize certain difficult samples, which causes better validation accuracy.

In the appendix (Appendix A.1, we dig deeper in the relationship between example forgetting and SGD instability by showing that removing frequently-forgotten samples decreases instability at the cost of validation accuracy.

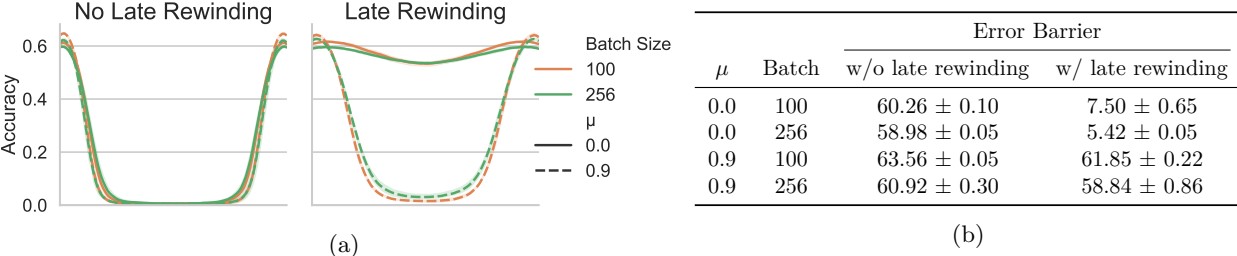

Figure 1: Instability for a dense ResNet-34 model trained on TinyImageNet with different training configurations. We show the full interpolation curve in **(a)** and the error barrier in **(b)**.

Table 1: Average forgetting events per sample, proportion of unforgettable samples and convergence statistics for different hyperparameter configurations used in training ResNet-34 models on TinyImageNet.

| $\mu$ | Batch Size | Forgetting events | Unforgettable % | Convergence |
|---|---|---|---|---|
| 0.0 | 100 | $34.73 \pm 12.54$ | $0.23 \pm 0.02\%$ | $0.907 \pm 0.003$ |
| 0.0 | 256 | $32.83 \pm 11.90$ | $0.27 \pm 0.02\%$ | $0.771 \pm 0.109$ |
| 0.9 | 100 | $42.78 \pm 14.37$ | $0.05 \pm 0.00\%$ | $0.773 \pm 0.042$ |
| 0.9 | 256 | $39.38 \pm 13.19$ | $0.11 \pm 0.02\%$ | $0.800 \pm 0.111$ |

**Convergence speed.** We can see that the approaches without momentum have a faster convergence than the approaches with momentum. This further highlights that while training with momentum results in a higher final accuracy, this is due to a more exploratory learning approach, as it takes significantly more training epochs for the momentum-based approaches to attain close to their final accuracy.

### 4.2 Impact on Lottery Tickets

In Table 2, we have listed the validation accuracy of Lottery Tickets found under different hyperparameter configurations at different sparsity levels (after every 5 pruning steps). In the next paragraphs, we use these results to highlight how parameter selection and the associated SGD instability of the training procedure impacts the resulting Lottery Tickets.

**Winning Tickets can be found without pretraining.** We notice that winning tickets (underlined entries in Table 2) exist in the configuration with $\mu = 0.0$, Batch Size $= 256$. This configuration was noted as having the lowest error barrier in Figure 1, which leads us to believe that the application of hyperparameters with lower error barriers can lead to the extraction of winning tickets. An important stipulation with these hyperparameters is that – such as already determined in the previous experiments – these have a slight negative effect on the generalization accuracy of the dense network. During the early pruning iterations (up until 73.79% sparsity for this configuration), this results in tickets that underperform w.r.t. their counterparts with less unstable hyperparameters. However, after this cutoff, we can see that the stable approach outperforms all other configurations without pretraining.

**Instability in relation to ticket performance.** While the error barriers for each hyperparameter configuration were relatively close in Figure 1, we notice a significant difference between the hyperparameter configurations w.r.t. ticket performances. Those that have lower SGD instability suffer from less accuracy degradation during the pruning process. This is best exemplified in the approach with $\mu = 0.9$, Batch Size $= 100$, which suffers from non-convergence at several sparsities for both approaches with and without pretraining. This shows that SGD instability is a significant factor in accuracy degradation via pruning, and that techniques such as pretraining, or smart hyperparameter selection are required to mitigate this instability. In the appendix in Appendix B we do additional experiments on the SGD instability of Lottery Tickets.

In Appendix C of the appendix, we show results for other dataset and network combinations, where we can make similar observations w.r.t. instability, existence of winning tickets, and ticket performance.

Table 2: Lottery Ticket extracted from ResNet-34 on TinyImageNet performances for different training parameters at different sparsity levels. Winning tickets are underlined. Starred(*) entries encompass one or multiple runs in which the ticket did not converge and remained at random chance.

| Ticket Parameters | | | Sparsity | | | | |
|---|---|---|---|---|---|---|---|
| $\mu$ | Batch Size | Pretrained | 0.00% | 67.23% | 89.26% | 96.48% | 98.85% |
| 0.0 | 100 | | 61.14% | 59.76% | 57.79% | 54.47% | 50.98% |
| 0.0 | 256 | ✗ | 59.33% | 59.95% | **60.20%** | **58.89%** | **57.09%** |
| 0.9 | 100 | | **63.74%** | **62.46%** | 35.95%* | 35.69%* | 15.09%* |
| 0.9 | 256 | | 61.57% | 60.37% | 56.93% | 52.76% | 47.34% |
| 0.0 | 100 | | 60.91% | 62.90% | **62.67%** | **60.37%** | 56.11% |
| 0.0 | 256 | ✓ | 59.27% | 60.01% | 60.08% | 58.59% | **56.40%** |
| 0.9 | 100 | | **63.45%** | **63.02%** | 60.10% | 57.73% | 35.10%* |
| 0.9 | 256 | | 61.66% | 61.22% | 58.95% | 55.91% | 52.49% |

Table 3: AIMP results for different hyperparameter settings of ResNet-34 on TinyImageNet.

| Ticket Parameters | | | | Sparsity | | | |
|---|---|---|---|---|---|---|---|
| Budget | $\mu$ | Pretrained | Batch Size | 67.23% | 89.26% | 96.48% | 98.85% |
| 20eps | 0.0 | ✗ | 256 | 58.90% | 56.95% | 54.13% | 49.73% |
| | 0.9 | ✗ | 100 | **62.82%** | **61.41%** | **58.49%** | **53.65%** |
| 50eps | 0.0 | ✗ | 256 | 58.74% | 56.75% | 53.22% | **49.58%** |
| | 0.9 | ✗ | 100 | **63.02%** | **59.51%** | **54.04%** | 39.04% |

### 4.3 Limiting Mask Search budget

In each iteration of *Mask Search*, the network is trained for a number of epochs, during which SGD noise accumulates, before the network is pruned and rewound. As such, we could expect that limiting the number of epochs trained in this phase, can limit the accuracy degradation incurred by these tickets. In fact, it has been observed by Zullich et al. (2021) that it is possible to limit the number of epochs in the *Mask Search* phase without significant impact on the resulting ticket performance, which they term Accelerated Iterative Magnitude Pruning (AIMP). We study this finding on more complex settings, with computational budgets of 20 and 50 epochs in Table 3, which represents 10% and 25%, respectively, of the original budget. We additionally modify the LR schedule to reflect upon this limited budget.

We notice that the tickets found under more unstable hyperparameters ($\mu = 0.9$, BS=100) via AIMP with a budget of 20eps outperform those found with the full budget via IMP. However, once we increase the budget to 50eps we notice a degradation in accuracy w.r.t. solutions found with 20eps. This shows that for unstable hyperparameters it is beneficial to limit the *Mask Search* phase, as a too high budget leads to too much SGD noise accumulation, and underperforming lottery tickets. Interestingly, for the more stable hyperparameters, we observe results we would expect, where limiting the budget leads to underperformance w.r.t. using the full budget, showing that AIMP primarily helps in settings with more unstable hyperparameters. In Appendix D of the appendix we corroborate these results with additional experiments on ResNet-18 + CIFAR-10.

### 4.4 Disentangling Mask Search and Sparse Training

Previous experiments have used the same hyperparameters for the *Mask Search* and *Sparse Training* phases. This follows common practice, and allows for a strict definition of 'winning' lottery tickets. However, we can also relax the definition slightly, where we still enforce the same number of training iterations, but allow for different hyperparameters between both phases. This allows us to disentangle the effects of hyperparameters in both phases on the final validation accuracy. To do this, we consider two tickets found with different

Table 4: Grid search over different Sparse Training parameters (indicated with a ‡), while keeping the Mask Search parameters (indicated with a †) fixed for ResNet-34 + TinyImageNet Lottery Tickets. Starred ($^*$) entries encompass one or multiple runs in which the ticket did not converge and remained at random chance.

| Ticket parameters | | | | | Sparsity | | | |
|---|---|---|---|---|---|---|---|---|
| $\mu^\dagger$ | Batch Size$^\dagger$ | Pretrained | $\mu^\ddagger$ | Batch Size$^\ddagger$ | 67.23% | 89.26% | 96.48% | 98.85% |
| 0.0 | 256 | ✗ | 0.0 | 100 | 58.48% | 57.07% | 53.47% | 50.02% |
| | | | 0.0 | 256 | 59.95% | **60.20%** | **59.98%** | **57.09%** |
| | | | 0.9 | 100 | **60.93%** | 59.21% | 57.55% | 54.31% |
| | | | 0.9 | 256 | 60.58% | 58.88% | 57.19% | 53.82% |
| 0.9 | 100 | ✗ | 0.0 | 100 | 58.36% | 51.41% | 48.35% | **42.86%** |
| | | | 0.0 | 256 | 58.30% | 51.47% | 48.48% | **42.86%** |
| | | | 0.9 | 100 | **62.46%** | 35.95%$^*$ | 35.69%$^*$ | 15.09%$^*$ |
| | | | 0.9 | 256 | 61.52% | **53.65%** | **51.34%** | 14.94%$^*$ |

hyperparameter configurations in *Mask Search*, and do a grid search on the hyperparameters used in *Sparse Training* in Table 4. First and foremost, we notice that for all tickets with a higher sparsity ($>67.23\%$), the *Mask Search* parameters are more influential in the final accuracy than the *Sparse Training* parameters. This is made clear as for those sparsities, no matter which hyperparameters are selected for *Sparse Training*, the tickets found with $\mu = 0.9$, Batch Size 256 in *Mask Search* show a significant accuracy increase over the other setting. For the low-sparsity case (67.23%), the hyperparameters which result in the best dense network performance in Table 2, also result in the best ticket performance when applied during *Sparse Training*, however these hyperparameters lose effectiveness for sparser tickets. These results show that the application of more stable hyperparameters throughout the *Mask Search* phase has a more significant impact on the trainability of Lottery Tickets compared to the hyperparameters used during *Sparse Training*.

## 5 Consequences of stability in Lottery Tickets

### 5.1 Untrained tickets can encode useful features

Earlier work by Zhou et al. (2019) likened the Mask Search procedure to training a model. They determined that it is possible to generate a mask for a of set of network weights, such that the network achieves better than random chance predictions, effectively outperforming a randomly initialized network. Such masks are called supermasks, and have been extracted for simple convolutional models on CIFAR-10, MNIST. Further accuracy gains for these supermasks could be achieved by directly optimizing the masks via gradient descent, which can be likened to the Strong Lottery Ticket Hypothesis (Ramanujan et al., 2020) where a mask is optimized for a set of weights to approximate the performance a fully trained network.

The phenomenon of supermasks is quite restrictive, as it assumes that by pruning; *(i)* useful features emerge within the feature extractor, and *(ii)* the classification layer can exploit those features. All of this should happen without modifying a single weight. In practice, during the earlier experiments we have not observed any ticket at meaningful sparsity that exhibited better than random chance without training.

Instead, we focus only on case *(i)*, meaning we still allow the training of a classification layer, but keep the pruned feature extractor frozen. Furthermore, we consider feature depth as well. Intuitively, if useful features are encoded within the ticket, then it should follow that the features in the deeper layers of the network are more expressive than those in the earlier layers, as is the case with a dense network. To determine this, we devise a linear evaluation experiment inspired by linear probing (Alain & Bengio, 2017).

**Linear feature evaluation.** After each ResNet block in the ticket we insert a linear probe, which is a channel-wise pooling operation followed by a linear classification layer. While the linear probe is training, we freeze all other parameters in the network, such that the resulting accuracy correctly reflects the predictive quality of the features. For consistency between different parameter configurations, we use a single set of

Table 5: Linear probing results at different locations in a ResNet-34 ticket extracted for TinyImageNet under different training configurations. Starred(*) entries correspond to dense features, and as such they have a sparsity of 0.00%.

| Ticket parameters | | | 67.23% sparsity | | 89.26% sparsity | |
|---|---|---|---|---|---|---|
| $\mu$ | Batch Size | Pretrained | Block 4 | Block 8 | Block 4 | Block 8 |
| 0.0 | 100 | | $4.26 \pm 0.39\%$ | $3.83 \pm 0.19\%$ | $3.97 \pm 0.25\%$ | $3.84 \pm 0.27\%$ |
| 0.0 | 256 | ✗ | $\mathbf{4.38 \pm 0.35\%}$ | $\mathbf{4.53 \pm 0.47\%}$ | $\mathbf{4.86 \pm 0.26\%}$ | $\mathbf{5.53 \pm 0.70\%}$ |
| 0.9 | 100 | | $1.96 \pm 0.46\%$ | $1.89 \pm 0.27\%$ | $1.29 \pm 0.40\%$ | $1.59 \pm 0.40\%$ |
| 0.9 | 256 | | $2.92 \pm 0.48\%$ | $2.63 \pm 0.53\%$ | $2.61 \pm 1.20\%$ | $2.04 \pm 0.90\%$ |
| 0.0 | 100 | | $6.66 \pm 0.42\%$ | $7.90 \pm 1.38\%$ | $\mathbf{6.39 \pm 0.41\%}$ | $7.24 \pm 1.52\%$ |
| 0.0 | 256 | ✓ | $\mathbf{6.67 \pm 0.30\%}$ | $\mathbf{9.55 \pm 0.18\%}$ | $6.32 \pm 0.29\%$ | $\mathbf{9.58 \pm 0.42\%}$ |
| 0.9 | 100 | | $4.55 \pm 0.15\%$ | $4.86 \pm 0.46\%$ | $4.23 \pm 0.08\%$ | $4.45 \pm 0.30\%$ |
| 0.9 | 256 | | $4.47 \pm 0.21\%$ | $4.53 \pm 0.33\%$ | $4.53 \pm 0.12\%$ | $4.73 \pm 0.12\%$ |
| Permuted ($\mu = 0.0$, BS=256) | | | $3.74 \pm 0.12\%$ | $3.39 \pm 0.09\%$ | $4.07 \pm 0.09\%$ | $4.09 \pm 0.16\%$ |
| Random dense network* | | | $3.66 \pm 0.82\%$ | $3.13 \pm 0.45\%$ | $3.66 \pm 0.82\%$ | $3.13 \pm 0.45\%$ |

hyperparameters to train the linear layer, as tests with different sets of hyperparameters only showed a minimal impact on the validation accuracy of the probe. We compare the results with a baseline, which consists of a ticket for which the mask is randomly layerwise permuted, and thus preserves the initialization and the layerwise sparsity.

In Table 5 we highlight the results at two sparsity levels, for two positions in the model (Blocks 4 & 8). The sparsity levels were chosen because of two reasons. First, they both correspond to the sparsity levels described in Table 2, and secondly at the 67.23% sparsity level, most tickets exhibit similar accuracy when fully trained. Block 4 is roughly equivalent to the middle of the model, while Block 8 considers the features at the last convolutional layer. We list the full results at all sparsities and blocks in Appendix E.1 of the appendix.

First we highlight that the randomly permuted tickets do not significantly outperform the features encoded in a random dense network. We see a slight increase in accuracy at higher sparsity, but no major impact. This shows that sparsity by itself does not promote the emergence of useful features, and in fact too much 'random' sparsity can inhibit good features. Secondly, we can divide the features in tickets in three main categories. The majority of the results show no significant increase or decrease w.r.t. the dense network. That being said, tickets found without pretraining but with $\mu = 0.9$ typically have lower quality features encoded, while those found with $\mu = 0.0$ and pretraining have much higher quality features. We notice that the stable configurations result in potent feature extractors at initialization. This shows that the mask search procedure can find useful features from initialization or weights in early training. Interestingly, in the unstable cases the features at the end of the network provide no accuracy gain compared to the intermediate features, which is opposite to that of the more stable tickets (late rewinding with $\mu = 0.0$) where we can notice a clear increase, and what we would expect in a trained network. A simple comparison shows that the remarkable performance of stable tickets can not be explained by those factors alone.

**Transferability of emergent features.** Previous research (Morcos et al., 2019; Mehta, 2019) has demonstrated that Lottery Tickets generalize well to other datasets. We further explore our earlier observations that usable features emerge in lottery tickets, and analyze their transferability to different datasets. In this experiment, we choose to transfer tickets extracted for CIFAR-10 as this dataset is the least complex of the considered datasets. This implicates that the features encoded by the mask should be less suited for more complex datasets. To enable transferability to datasets with different number of classes, we replace the linear layer with a new linear layer with a target-specific number of outputs, and freeze all other layers. While this procedure leads to a loss of sparsity in the classification layer, this is the only possible approach short of resparsifying the linear layer, which might induce side effects.

Table 6: Transferability of frozen ResNet-18 tickets extracted on CIFAR-10 with different configurations. Starred(*) entries correspond to dense trained networks, meaning the feature extractors have a sparsity of 0.00% and have been trained either on the source or the target dataset.

| Ticket parameters | | | Target dataset | | | |
|---|---|---|---|---|---|---|
| $\mu$ | Batch Size | Pretrained | MNIST | CIFAR-100 | TinyImageNet | EuroSAT |
| 0.0 | 100 | | $82.11 \pm 0.48\%$ | $16.66 \pm 0.20\%$ | $7.42 \pm 0.14\%$ | $70.46 \pm 0.94\%$ |
| 0.0 | 256 | ✗ | $\mathbf{97.25 \pm 0.15\%}$ | $\mathbf{40.57 \pm 1.07\%}$ | $\mathbf{18.13 \pm 0.33\%}$ | $\mathbf{85.21 \pm 0.56\%}$ |
| 0.9 | 100 | | $77.03 \pm 0.31\%$ | $12.73 \pm 0.82\%$ | $5.49 \pm 0.21\%$ | $61.00 \pm 1.81\%$ |
| 0.9 | 256 | | $77.30 \pm 1.60\%$ | $13.35 \pm 0.74\%$ | $6.28 \pm 0.92\%$ | $60.74 \pm 5.21\%$ |
| 0.0 | 100 | | $96.90 \pm 0.03\%$ | $36.18 \pm 1.02\%$ | $18.48 \pm 0.79\%$ | $82.40 \pm 0.98\%$ |
| 0.0 | 256 | ✓ | $\mathbf{97.45 \pm 0.21\%}$ | $\mathbf{41.63 \pm 0.91\%}$ | $\mathbf{21.56 \pm 1.21\%}$ | $\mathbf{85.47 \pm 0.98\%}$ |
| 0.9 | 100 | | $90.85 \pm 1.60\%$ | $17.34 \pm 1.80\%$ | $8.46 \pm 0.29\%$ | $69.37 \pm 1.06\%$ |
| 0.9 | 256 | | $92.79 \pm 2.67\%$ | $18.96 \pm 4.22\%$ | $10.47 \pm 1.91\%$ | $70.86 \pm 1.43\%$ |
| Frozen dense* ($\mu = 0.0$, BS=256) | | | $95.55 \pm 0.15\%$ | $39.82 \pm 0.12\%$ | $12.46 \pm 0.20\%$ | $78.59 \pm 0.81\%$ |
| Frozen dense* ($\mu = 0.9$, BS=100) | | | $71.17 \pm 0.81\%$ | $19.15 \pm 0.22\%$ | $4.41 \pm 0.25\%$ | $56.42 \pm 3.54\%$ |
| Retrained dense network* | | | $99.58 \pm 0.04\%$ | $77.40 \pm 0.37\%$ | $60.15 \pm 0.00\%$ | $98.51 \pm 0.53\%$ |

Transferring is done to MNIST (LeCun et al., 2010), CIFAR-100 (Krizhevsky et al., 2009), TinyImageNet (Le & Yang, 2015) and EuroSAT (Helber et al., 2019). The reasoning for these datasets is as follows. MNIST is an easy dataset, albeit monochrome, which features different instances from CIFAR-10 (numbers, rather than objects). CIFAR-100 has been generated using the same process as CIFAR-10, but is more complex and features mutually exclusive images and classes. TinyImageNet is an even more challenging dataset, with more classes, less instances per class, and a higher resolution than CIFAR-10. Finally, EuroSAT features higher resolution landscape images, rather than object images. Each of these datasets has characteristics not present in CIFAR-10, which serves to demonstrate the versatility of the features present within the lottery ticket. During the network transfer, we normalize the images with the mean and standard deviation, and rescale to 32×32.

We show the results for 89.26% sparsity tickets in Table 6, where we additionally compare with several baselines. We finetune dense networks under several hyperparameter regimes to show the transferability of features specialized for a different dataset in unstable and less unstable regimes. We also provide an upper limit on the model performance by training a dense network from scratch on the target dataset.

We observe that the features learned in high instability regimes are significantly less transferable to other datasets, than those learned in more stable regimes. This is the case for both the dense network and also the derived Lottery Tickets, showing that this property is likely inherited by the ticket from the dense network. That being said, in no circumstances does transferring beat the retraining performance. Whether the ticket outperforms the dense network or not seems to be dependent on the dataset and instability. Stable tickets always outperform the dense network, but the most unstable tickets only do for certain datasets. In the appendix (Appendix E.2), we transfer CIFAR-100 and TinyImageNet tickets, and notice that while the validation accuracy of the transferred ticket is still remarkable, we fail to consistently outperform the features in the trained dense network. This is of course not that unexpected, seeing that 1) the tickets we transfer are still frozen at initialization and encode worse features than those of a dense network, as measured by our linear probes earlier, and 2) the features in the dense network are learned for a more complex setting and as such are already better transferable to less complex datasets.

## 5.2 Training tickets with limited data

Having observed this emergence of useful features to certain lottery tickets, we next want to determine how we can further exploit this. For this we consider accelerating the Sparse Training phase. Rather than using the full dataset, which can be costly and cannot be run on small devices, instead we consider few-shot learning setting, where a limited number of samples is used.

Table 7: Validation accuracies of a 89.26% sparse ResNet-34 ticket when trained on a TinyImageNet subset.

| | Ticket parameters | | Subset sizes | | | |
|---|---|---|---|---|---|---|
| $\mu$ | Batch Size | Pretrained | 1% | 2% | 5% | 10% |
| 0.0 | 100 | | $4.72 \pm 0.46\%$ | $7.69 \pm 0.39\%$ | $16.46 \pm 0.24\%$ | $27.34 \pm 0.18\%$ |
| 0.0 | 256 | ✗ | $\mathbf{10.49 \pm 1.14\%}$ | $\mathbf{17.90 \pm 1.72\%}$ | $\mathbf{29.51 \pm 1.71\%}$ | $\mathbf{37.87 \pm 0.43\%}$ |
| 0.9 | 100 | | $3.41 \pm 0.28\%$ | $5.18 \pm 0.59\%$ | $14.33 \pm 0.55\%$ | $8.36 \pm 13.61\%$ |
| 0.9 | 256 | | $3.40 \pm 0.62\%$ | $6.17 \pm 0.85\%$ | $11.42 \pm 0.31\%$ | $21.25 \pm 1.63\%$ |
| 0.0 | 100 | | $10.31 \pm 1.19\%$ | $15.29 \pm 0.74\%$ | $26.01 \pm 1.19\%$ | $36.35 \pm 1.14\%$ |
| 0.0 | 256 | ✓ | $\mathbf{15.94 \pm 0.69\%}$ | $\mathbf{22.90 \pm 1.21\%}$ | $\mathbf{34.18 \pm 0.64\%}$ | $\mathbf{43.40 \pm 0.26\%}$ |
| 0.9 | 100 | | $6.00 \pm 0.58\%$ | $8.76 \pm 0.86\%$ | $17.59 \pm 1.09\%$ | $26.50 \pm 0.83\%$ |
| 0.9 | 256 | | $4.95 \pm 0.24\%$ | $7.65 \pm 0.78\%$ | $15.31 \pm 0.75\%$ | $25.55 \pm 0.61\%$ |
| | *Dense Network* | | $4.32 \pm 0.13\%$ | $6.33 \pm 0.19\%$ | $13.19 \pm 0.33\%$ | $25.13 \pm 0.57\%$ |

In our experiments, we use the following subset sizes: [1% , 2%, 5%, 10%]. As we focus on small dataset sizes, we will use the random selection method to select class-balanced subsets, since this has been shown to work best in those cases (Guo et al., 2022). To allow for comparability between TinyImageNet and CIFAR-10, we have chosen 1% as a bottom limit, as TinyImageNet features 500 images per class, while CIFAR-10 features 5000 images per class. Going lower than 1% could be feasible for CIFAR-10, but will be difficult for TinyImageNet, as then the randomness of the subset selection process will significantly impact the performance of the trained network. Results for tickets at 89.26% sparsity on TinyImageNet are listed in Table 7. Additional results and visualizations can be found in the appendix in Appendix F.

We notice that a significant portion of the validation accuracy attained with the full training dataset can be recovered in scenarios with higher SGD stability. In the most extreme case, we can recover $\sim 75\%$ validation accuracy of the full dataset by training with 50 randomly chosen images per class of TinyImageNet. This is a relative gain of $\sim 72\%$ over training the dense network with that subset. In Appendix F of the appendix, we show results for a more robust dataset subset selection criterion, for which the results follow a similar trend.

## 6 Discussion

### 6.1 Loss Basin interpretation

It has been observed that within the loss landscape there exists multiple basins of attraction that contain local optima. A neural network, when trained, converges to a basin of attraction and reaches a local optimum in that basin. Typically this final loss basin for the network is determined early in training, as shown in Fort et al. (2020), while in the rest of the training phase the optimizer will converge to that loss basin. By applying pretraining and rewinding, the aim is to select a loss basin to which all subsequent tickets will converge. To determine whether two networks lie in the same loss basin, we can calculate the error barrier across the linear interpolation. As such, the definition of stability to SGD noise as introduced by Frankle et al. (2020) simply indicates whether a network when trained under different sets of SGD noise converges to the same loss basin. The time required to determine these loss basins differs significantly for different hyperparameter configurations (see Figure 1, and Appendix C.3).

Dense networks that are stable to SGD noise (always converge to the same loss basin) also have the property that derived lottery tickets are also stable (Frankle et al., 2020). Even more is that in those cases subsequent tickets also lie in the same loss basin, as they can be connected via a linear path without a significant error barrier (Paul et al., 2023). However, in our experiments we have found winning lottery tickets without pretraining for which the dense network is unstable to SGD noise ($\mu = 0.0$, Batch Size = 256), directly contradicting the common knowledge. What we notice however in Figure 2 is that for those hyperparameters some of the subsequent tickets lie in the same basin, albeit starting from a significant sparsity, rather than

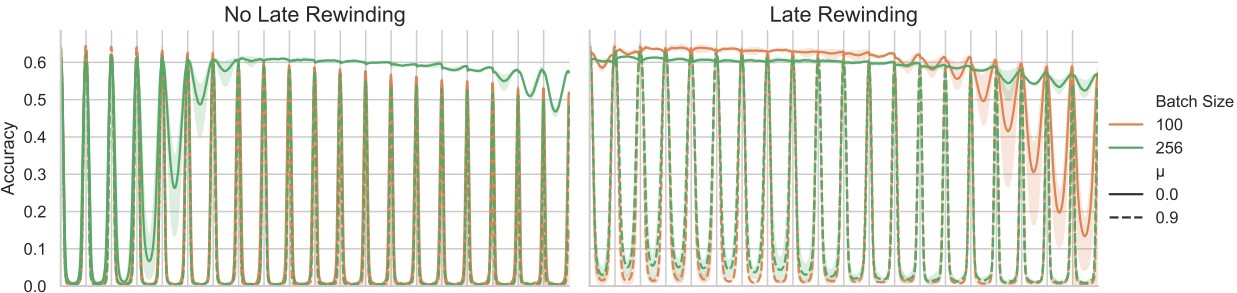

Figure 2: Error barrier between trained tickets found in subsequent pruning iterations for ResNet-34 trained on TinyImageNet. Gray vertical lines indicate the trained tickets.

directly pruning. This indicates that while initially the tickets converge to different loss basins, somehow the loss basins either merge together when reducing the dimensionality as the error barrier diminished throughout pruning, or one of the loss basins disappears. Additional results are listed in Appendix B.2.

**The impact of momentum.** The application of momentum allows for higher validation accuracy for the dense network, by compounding gradients over different minibatches. However, this clearly has an impact on the SGD stability of the network. We can see from the forgetting scores Table 1 that momentum causes a significant increase in forgetting events, meaning a more exploratory optimization path, and going through different loss basins. This finally results in a better loss basin that is found, as we have a higher validation accuracy for the model, but we hypothesize that this basin is much narrower than the basins found without momentum. As such, after pruning this loss basin becomes more difficult to find, leading to more significant accuracy degradation when coupled with the exploratory behaviour. In the appendix (Appendix C.3) we show empirically that solutions with $\mu = 0.9$ take significantly longer training to select a loss basin by calculating the optimal rewind point.

**Small Batch Size.** When using a small batch size, the gradient of a batch is more different from the average gradient, as the individual samples have a higher relative contribution. As such, the optimization process is more noisy and influenced more by the difficult samples which can lead the model to better loss basins. When a lot of parameters are removed, it is more difficult for the model to fit the difficult samples, and this can cause accuracy loss.

**Limited training.** By limiting the training we also limit the exploration time. In the cases with momentum this means that rather than continuing to explore for the best possible loss basin, instead the model converges to a 'good enough' basin. This avoids issues where the loss basin is not attainable anymore after pruning. Conversely, in cases without momentum, the loss basin is attained early during training.

## 6.2 Usefulness of SGD instability metric

While a dense network with a low error barrier, i.e., that converges to the same loss basin when trained with different sets of SGD noise, is an accurate indicator of the existence of winning tickets within that network, the inverse is not always true. In our experiments (Figure 1, and Table 2) we have noticed that winning tickets can exist in dense networks where the error barrier is higher than the 'stable' threshold, or even virtually identical to that of dense networks where no winning tickets exists. Rather what we consistently notice is that in dense networks that produce winning tickets, the model becomes stable to SGD noise relatively early during training (see Appendix C.3), and that at some point during the pruning of the network the tickets do become stable, rather than the dense network being stable. A better indication for the existence of winning tickets could be the amount of training required for a dense network to become stable to SGD noise, but this is a significantly more expensive metric to calculate.

### 6.3 The emergence of useful features

We notice in several experiments that, when instability is low enough, the tickets found via IMP contain useful features without additional training. This suggests that by pruning repeatedly some information of the dataset is encoded within the mask, dependent on how low SGD instability is. This information can be used to recover a nontrivial percentage (more than 15% in the case of TinyImageNet) of the validation accuracy of a fully trained dense network, by finetuning a linear classifier on the features.

This effect has been shown to be independent of the application of late-rewinding – which encodes some information in the weights by virtue of pretraining. Additionally, the features are not overfitting the dataset, but can rather be transferred to other datasets, and can even outperform those encoded in dense networks trained on the source dataset when transferred in several cases.

This behavior is reminiscent of the Strong Lottery Ticket Hypothesis (Ramanujan et al., 2020), which poses that any trained network can be approximated by pruning connections from a sufficiently large untrained network. There is however a nuanced difference, in that the stable tickets found by our approach require a finetuned classification layer to achieve good validation accuracy.

### 6.4 Other causes of SGD instability

While we limit our research to these two hyperparameters and study those in more detail, these are certainly not the only influences on SGD instability. Other notable factors include but are not limited to data shuffling during training, and data augmentation. Opposite to those factors, batch size and momentum are much less treated as a given during training.

## 7 Conclusion

By studying different values for the momentum and batch size hyperparameters for the training of a dense network, we notice that values which have positive effect on the generalization of dense networks, such as a high momentum and lower batch size – as already reported in the literature, result in higher instability to SGD noise.

When applying these insights to the Lottery Ticket Hypothesis, we notice that with certain hyperparameter configurations we can skip the pretraining step and still find winning tickets in complex scenarios, opposite to previous assumptions in Frankle et al. (2020). While these tickets do not reach optimal validation accuracy at lower sparsities (<67.23% sparsity), we show that they outperform all other tested hyperparameter configurations at higher sparsities, as they feature less accuracy degradation throughout pruning. This difference in accuracy degradation is even present when considering pretraining.

On top of exhibiting the winning phenomenon, and the higher validation accuracy at deeper sparsities, we show that tickets found with a higher stability exhibit several desirable properties encoded by the pruned initialization. We notice that these tickets achieve better few-shot generalizability, and can in fact be used as frozen feature extractors to a remarkable accuracy.

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

## Appendices

- Appendix A contains additional results for experiments conducted in Section 4.1 of the main paper. More specifically interpolation curves for the different different dataset and network combinations, and visualizations of the Train Convergence at different sparsities for tickets are provided. To complement these results, we highlight the relationship between forgetting events and Linear Mode Connectivity in Appendix A.1.

- Appendix B contains stability results for Lottery Tickets, where we repeat the experiments done in Section 4.1 on Lottery Tickets, rather than dense networks. Additionally, we highlight the error barriers between different lottery tickets, to determine whether subsequent lottery tickets lie in the same loss basin in Appendix B.2.

- Appendix C contains the full results for the experiments conducted in Section 4.2. This includes visualizations of the evolution of ticket validation accuracy for each setting, as well as tabular representations for the ResNet-18 + {CIFAR-10, CIFAR-100} setting. More extreme batch sizes for CIFAR-10 are explored in Appendix C.1, additional momentum values are explored in Appendix C.2, and the optimal rewind points for different configurations are calculated in Appendix C.3.

- Appendix D contains additional results for the experiments conducted in Section 4.3. We show additional results on ResNet-18 + CIFAR-10 for the same reduced computational budgets.

- Appendix E contains additional results for the experiments conducted in Section 5.1. More specifically, we show transferability for other datasets (Appendix E.2), and visualize the evolution of probe accuracy in function of sparsity and network depth for the different settings (Appendix E.1).

- Appendix F contains additional results for the experiments conducted in Section 5.2. These include results for other settings, and another subset selection criterion by using sample difficulty from Toneva et al. (2019) (Appendix F.1).

- Appendix G contains the hyperparameter configurations used in the experiments.

## A  Additional Training Stability Results

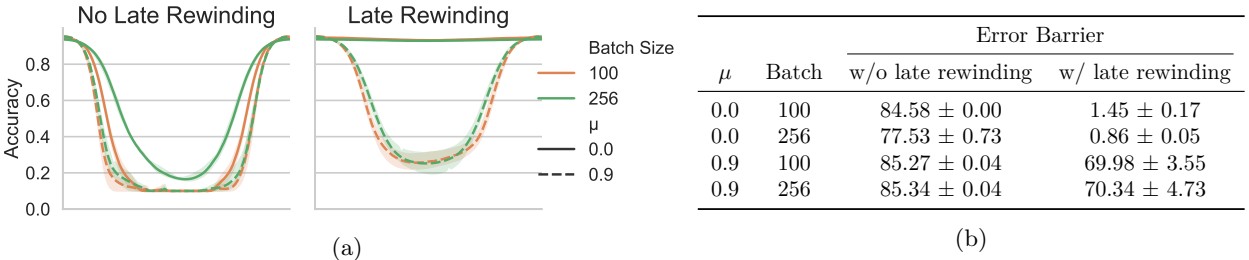

| | | Error Barrier | |
|---|---|---|---|
| $\mu$ | Batch | w/o late rewinding | w/ late rewinding |
| 0.0 | 100 | $84.58 \pm 0.00$ | $1.45 \pm 0.17$ |
| 0.0 | 256 | $77.53 \pm 0.73$ | $0.86 \pm 0.05$ |
| 0.9 | 100 | $85.27 \pm 0.04$ | $69.98 \pm 3.55$ |
| 0.9 | 256 | $85.34 \pm 0.04$ | $70.34 \pm 4.73$ |

(a)                                        (b)

Figure 3: Instability for a dense **ResNet-18** model trained on **CIFAR10** with different training configurations. We show the full interpolation curve in **(a)** and the error barrier in **(b)**. Notice that the interpolation error with $\mu = 0.0$, 256 BS is higher than random chance, which we do not observe for other dataset & network combinations.

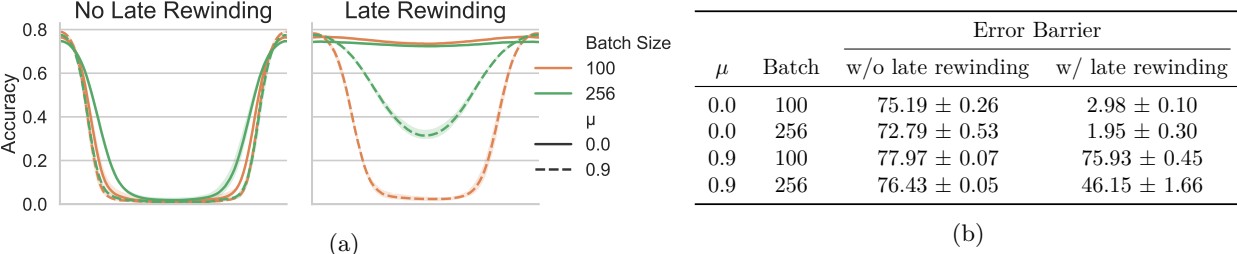

(a)

| | | Error Barrier | |
|---|---|---|---|
| $\mu$ | Batch | w/o late rewinding | w/ late rewinding |
| 0.0 | 100 | $75.19 \pm 0.26$ | $2.98 \pm 0.10$ |
| 0.0 | 256 | $72.79 \pm 0.53$ | $1.95 \pm 0.30$ |
| 0.9 | 100 | $77.97 \pm 0.07$ | $75.93 \pm 0.45$ |
| 0.9 | 256 | $76.43 \pm 0.05$ | $46.15 \pm 1.66$ |

(b)

Figure 4: Instability for a dense **ResNet-18** model trained on **CIFAR100** with different training configurations. We show the full interpolation curve in **(a)** and the error barrier in **(b)**.

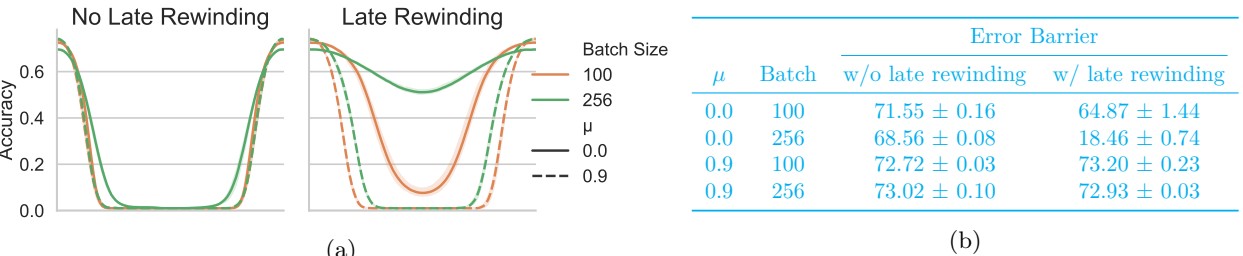

(a)

| | | Error Barrier | |
|---|---|---|---|
| $\mu$ | Batch | w/o late rewinding | w/ late rewinding |
| 0.0 | 100 | $71.55 \pm 0.16$ | $64.87 \pm 1.44$ |
| 0.0 | 256 | $68.56 \pm 0.08$ | $18.46 \pm 0.74$ |
| 0.9 | 100 | $72.72 \pm 0.03$ | $73.20 \pm 0.23$ |
| 0.9 | 256 | $73.02 \pm 0.10$ | $72.93 \pm 0.03$ |

(b)

Figure 5: Instability for a dense **VGG16** model trained on **CIFAR100** with different training configurations. We show the full interpolation curve in **(a)** and the error barrier in **(b)**.

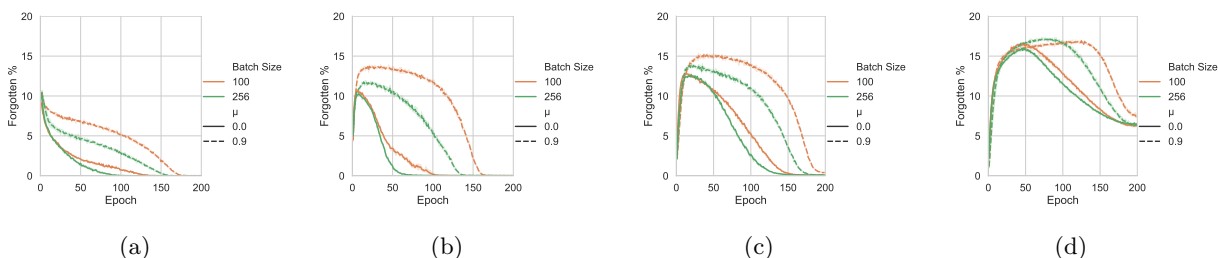

(a)      (b)      (c)      (d)

Figure 6: The evolution of forgetting events during training for **(a)** ResNet-18 + CIFAR-10, **(b)** ResNet-18 + CIFAR-100, **(c)** VGG16 + CIFAR-100, and **(d)** ResNet-34 + TinyImageNet given different hyperparameter configurations. Notice that approaches with $\mu = 0.9$ consistently have significantly more forgetting events late in training.

Table 8: Average forgetting events per sample, proportion of unforgettable samples and convergence statistics for different hyperparameter configurations used in training **ResNet-18** models on **CIFAR-10**.

| $\mu$ | Batch Size | Forgetting events | Unforgettable % | Convergence |
|---|---|---|---|---|
| 0.0 | 100 | $5.73 \pm 8.26$ | $42.90 \pm 0.16\%$ | $0.960 \pm 0.001$ |
| 0.0 | 256 | $3.35 \pm 4.44$ | $42.18 \pm 1.18\%$ | $0.939 \pm 0.003$ |
| 0.9 | 100 | $13.43 \pm 15.79$ | $22.92 \pm 0.50\%$ | $0.903 \pm 0.003$ |
| 0.9 | 256 | $8.61 \pm 11.23$ | $27.68 \pm 3.05\%$ | $0.927 \pm 0.005$ |

Table 9: Average forgetting events per sample, proportion of unforgettable samples and convergence statistics for different hyperparameter configurations used in training **ResNet-18** models on **CIFAR-100**.

| $\mu$ | Batch Size | Forgetting events | Unforgettable % | Convergence |
|------|------|------|------|------|
| 0.0 | 100 | $6.78 \pm 5.44$ | $16.21 \pm 0.19\%$ | $0.927 \pm 0.002$ |
| 0.0 | 256 | $5.10 \pm 4.06$ | $17.15 \pm 0.23\%$ | $0.905 \pm 0.003$ |
| 0.9 | 100 | $25.46 \pm 15.62$ | $4.59 \pm 0.15\%$ | $0.814 \pm 0.001$ |
| 0.9 | 256 | $15.77 \pm 11.57$ | $8.78 \pm 0.20\%$ | $0.862 \pm 0.004$ |

Table 10: Average forgetting events per sample, proportion of unforgettable samples and convergence statistics for different hyperparameter configurations used in training **VGG16** models on **CIFAR-100**.

| $\mu$ | Batch Size | Forgetting events | Unforgettable % | Convergence |
|------|------|------|------|------|
| 0.0 | 100 | $16.48 \pm 10.03$ | $3.77 \pm 0.09\%$ | $0.884 \pm 0.002$ |
| 0.0 | 256 | $13.84 \pm 7.77$ | $2.99 \pm 0.11\%$ | $0.840 \pm 0.002$ |
| 0.9 | 100 | $32.68 \pm 13.75$ | $0.14 \pm 0.02\%$ | $0.708 \pm 0.003$ |
| 0.9 | 256 | $24.93 \pm 13.90$ | $1.70 \pm 0.04\%$ | $0.807 \pm 0.001$ |

## A.1 SGD instability and forgetting events

As shown in Toneva et al. (2019), a subset of the CIFAR-10 dataset, selected by removing samples with a low forgetting score, can be used to train a ResNet-18 model without loss of generalization, as compared to the full dataset. The authors show that this is true for removing up to the 30% of the samples with the lowest forgetting events. This indicates that the samples with more forgetting events are more critical for the model training. We hypothesize that these samples, while more critical to the performance, also serve as a factor of SGD noise. To verify this, we generate dataset subsets where we remove the samples with the least forgetting events (termed 'easiest'), and datasets where we remove the samples with the most forgetting events (termed 'hardest'). For each subset, we train a ResNet-18 model on CIFAR-10 with $\mu = 0.0$, BS = 256 for three random seeds. We then show the interpolation curves, error barriers and validation accuracies for each dataset subset in Figure 7.

From these results, we can confirm that removing the least-forgotten samples has no significant impact on the validation accuracy, but that removing the most-forgotten samples significantly impacts the generalization error. When analyzing the instability to SGD noise, we notice a decrease in instability when removing harder samples. Removing an equivalent amount of easy samples has no such significant impact. If anything, we can notice a slight worsening in the interpolation curve, as the error barrier is reached earlier during interpolation.

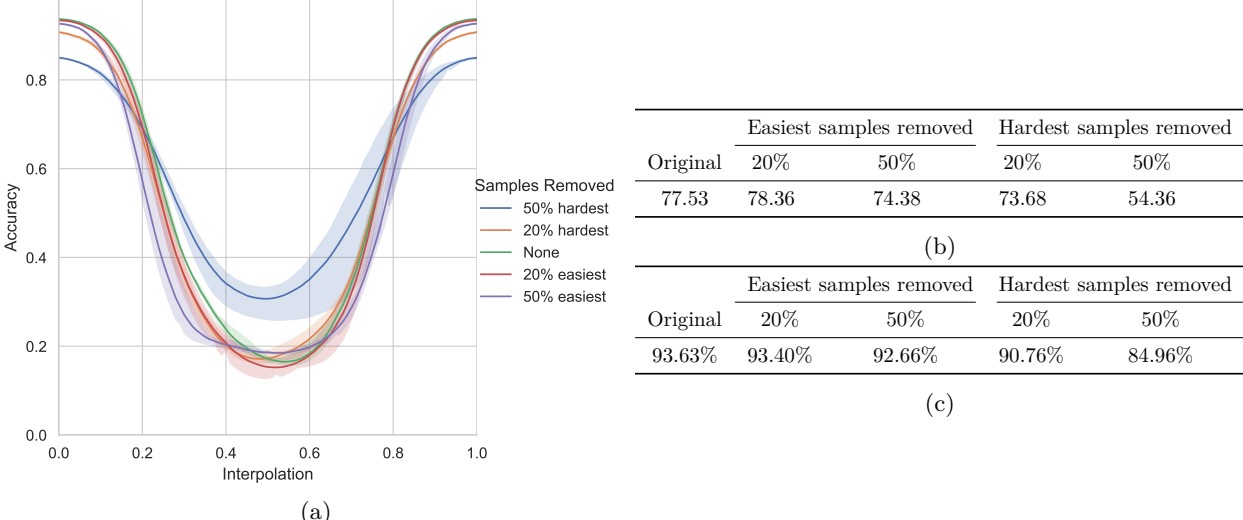

|  | Easiest samples removed | | Hardest samples removed | |
|---|---|---|---|---|
| Original | 20% | 50% | 20% | 50% |
| 77.53 | 78.36 | 74.38 | 73.68 | 54.36 |

(b)

|  | Easiest samples removed | | Hardest samples removed | |
|---|---|---|---|---|
| Original | 20% | 50% | 20% | 50% |
| 93.63% | 93.40% | 92.66% | 90.76% | 84.96% |

(c)

(a)

Figure 7: **(a)** Linear interpolation interpolation curves, **(b)** error barriers, and **(c)** validation accuracies of **ResNet18** networks trained on different subsets of **CIFAR10**. The networks are trained with $\mu = 0.0$, 256 batch size.

# B    Training stability in the Lottery Ticket Hypothesis

## B.1    Training stability of Lottery Tickets

In the main paper, as well as in Table 15, we have only considered the instability of dense networks to SGD noise. However, once the network is pruned, the dimensionality of the loss space is diminished, which constrains the optimization process and can have a significant impact on the instability of tickets to SGD noise. For this purpose, we calculate the error barrier for ResNet-34 lottery tickets on TinyImageNet at 67.23%, and 89.26% sparsity, and visualize the results in Figures 8 and 9.

When comparing these results with Figure 1 in the main paper, it is evident that lottery tickets have different instability from the dense network. We can see that without pretraining the configuration with $\mu = 0.0$, Batch Size=256 has a significantly lower error barrier compared to those of the dense network, showing that while this configuration is not as stable for the dense network, it is much more stable for the resulting tickets. Additionally, in the case of late-rewinding, we also see lower loss barriers for the settings without momentum ($\mu = 0.0$) w.r.t. those found for the dense network. The trend from these experiments is clear, that a smart hyperparameter selection can have significant impact on the stability of lottery tickets, even if its impact is less noticeable for the dense networks.

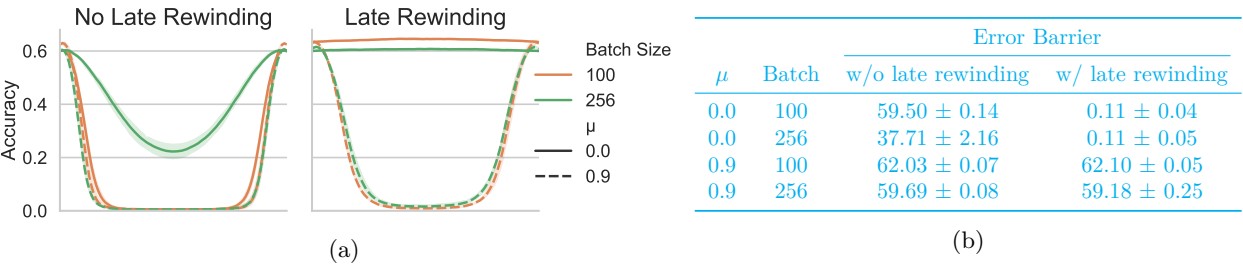

|  |  | Error Barrier | |
|---|---|---|---|
| $\mu$ | Batch | w/o late rewinding | w/ late rewinding |
| 0.0 | 100 | $59.50 \pm 0.14$ | $0.11 \pm 0.04$ |
| 0.0 | 256 | $37.71 \pm 2.16$ | $0.11 \pm 0.05$ |
| 0.9 | 100 | $62.03 \pm 0.07$ | $62.10 \pm 0.05$ |
| 0.9 | 256 | $59.69 \pm 0.08$ | $59.18 \pm 0.25$ |

(a)

(b)

Figure 8: Instability for a **ResNet-34 ticket at 67.23%** sparsity trained on **TinyImageNet** with different training configurations. We show the full interpolation curve in **(a)** and the error barrier in **(b)**.

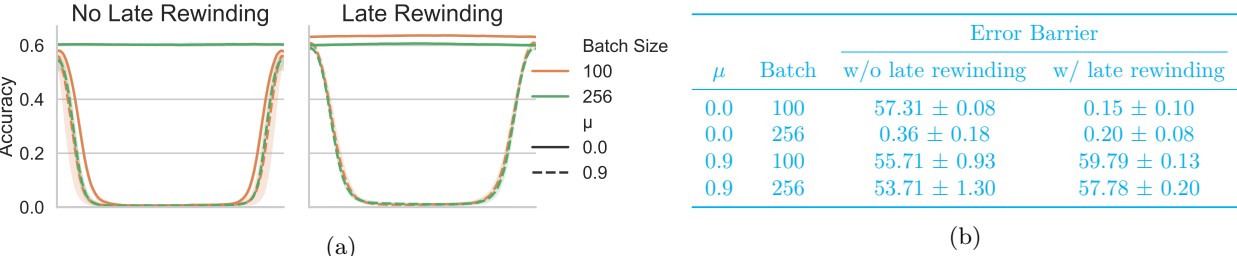

(a)                                                (b)

|  | | Error Barrier | |
| $\mu$ | Batch | w/o late rewinding | w/ late rewinding |
| 0.0 | 100 | $57.31 \pm 0.08$ | $0.15 \pm 0.10$ |
| 0.0 | 256 | $0.36 \pm 0.18$ | $0.20 \pm 0.08$ |
| 0.9 | 100 | $55.71 \pm 0.93$ | $59.79 \pm 0.13$ |
| 0.9 | 256 | $53.71 \pm 1.30$ | $57.78 \pm 0.20$ |

Figure 9: Instability for a **ResNet-34 ticket at 89.26%** sparsity trained on **TinyImageNet** with different training configurations. We show the full interpolation curve in **(a)** and the error barrier in **(b)**.

## B.2   Mode connectivity between subsequent tickets.

In the main paper we have shown that for ResNet-34 + TinyImageNet a configuration ($\mu = 0.0$, Batch Size=256) exists which results in winning tickets, while the dense network is not stable to SGD noise. Instead, what happens is that throughout the pruning process the resulting tickets become stable to SGD noise (as seen previously in this section. This has as a result that not only the tickets are stable to SGD noise, but also that the error barriers between the subsequent tickets vanish. Below, in Figures 10 to 12 we demonstrate that this occurrence is not unique to the ResNet-34 + TinyImageNet setting, but rather occurs in all tested settings, albeit the sparsity at which the subsequent tickets lie in the same loss basin differs wildly. The threshold at which this occurs is likely linked to the dataset and model combination, as we see different thresholds for different models.

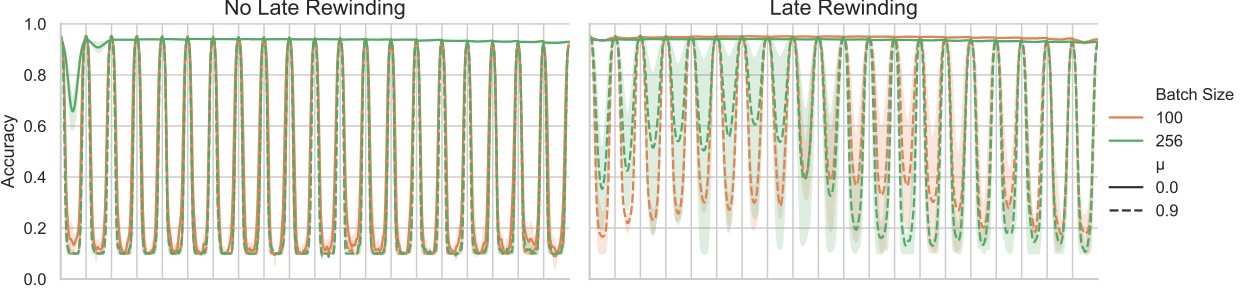

Figure 10: Error barrier between trained tickets found in subsequent pruning iterations for **ResNet-18** trained on **CIFAR-10**. Gray vertical lines indicate the trained tickets.

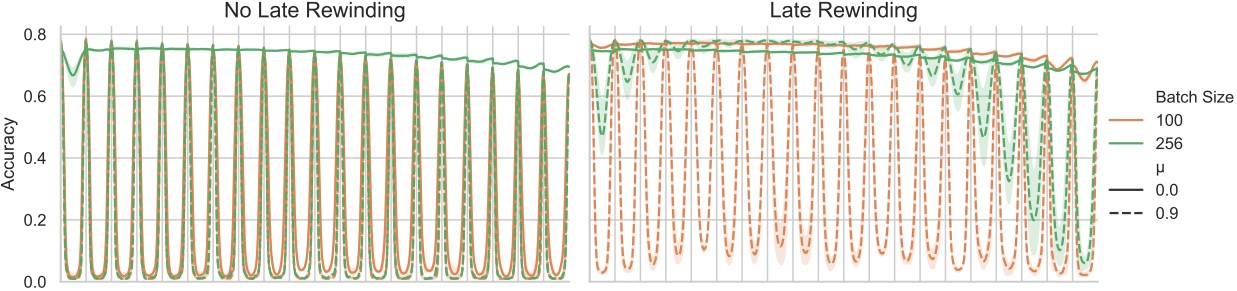

Figure 11: Error barrier between trained tickets found in subsequent pruning iterations for **ResNet-18** trained on **CIFAR-100**. Gray vertical lines indicate the trained tickets.

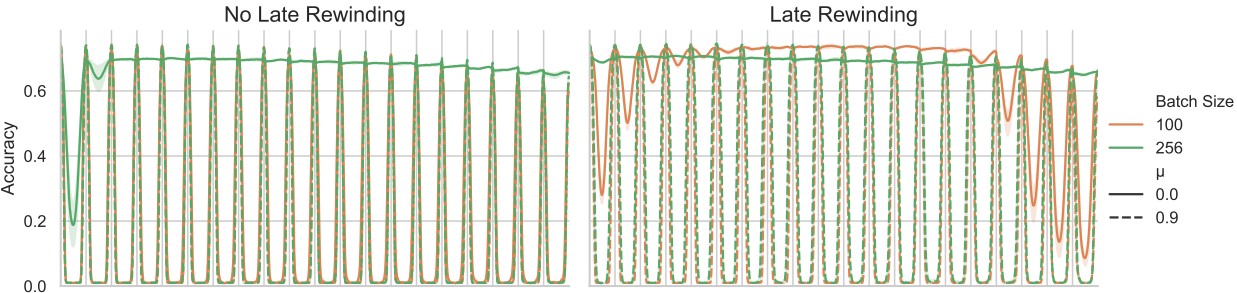

Figure 12: Error barrier between trained tickets found in subsequent pruning iterations for **VGG16** trained on **CIFAR-100**. Gray vertical lines indicate the trained tickets.

# C Full Lottery Ticket results

Table 11: Lottery Ticket extracted from **ResNet-18** on **CIFAR-10 / CIFAR-100** performances for different training parameters at different sparsity levels. Winning tickets are underlined.

| | Ticket Parameters | | | | Sparsity | | | |
|---|---|---|---|---|---|---|---|---|
| Dataset | $\mu$ | Batch Size | Pretrained | 0.00% | 67.23% | 89.26% | 96.48% | 98.85% |
| CIFAR-10 | 0.0 | 100 | ✗ | 94.58% | 94.39% | 93.93% | 92.95% | 91.10% |
| | 0.0 | 256 | | 93.63% | 94.02% | 93.93% | **93.39%** | **92.67%** |
| | 0.9 | 100 | | **95.24%** | **94.99%** | **94.39%** | 93.25% | 91.41% |
| | 0.9 | 256 | | 95.18% | 94.98% | 94.29% | 93.36% | 91.39% |
| | 0.0 | 100 | ✓ | 94.62% | 95.06% | **94.95%** | **94.71%** | **93.95%** |
| | 0.0 | 256 | | 93.54% | 93.97% | 93.94% | 93.50% | 92.77% |
| | 0.9 | 100 | | **95.30%** | 95.13% | 94.85% | 93.90% | 92.31% |
| | 0.9 | 256 | | 95.24% | **95.16%** | 94.55% | 93.57% | 91.86% |
| CIFAR-100 | 0.0 | 100 | ✗ | 76.66% | 75.57% | 73.53% | 70.24% | 66.43% |
| | 0.0 | 256 | | 74.39% | 75.03% | **74.46%** | **72.43%** | **68.65%** |
| | 0.9 | 100 | | **78.54%** | **76.60%** | 73.91% | 70.33% | 65.05% |
| | 0.9 | 256 | | 77.74% | 76.47% | 74.07% | 70.23% | 65.89% |
| | 0.0 | 100 | ✓ | 76.57% | 76.76% | 76.23% | 74.23% | **69.99%** |
| | 0.0 | 256 | | 74.41% | 74.62% | 73.54% | 71.76% | 67.93% |
| | 0.9 | 100 | | **78.18%** | 77.20% | 74.66% | 70.58% | 65.80% |
| | 0.9 | 256 | | 77.71% | **77.57%** | **77.24%** | **74.35%** | 69.21% |

Table 12: Lottery Ticket extracted from **VGG16** on **CIFAR-100** performances for different training parameters at different sparsity levels. Winning tickets are underlined.

| | Ticket Parameters | | | Sparsity | | | |
|---|---|---|---|---|---|---|---|
| $\mu$ | Batch Size | Pretrained | 0.00% | 67.23% | 89.26% | 96.48% | 98.85% |
| 0.0 | 100 | | 72.77% | 71.60% | 69.81% | 66.37% | 60.91% |
| 0.0 | 256 | ✗ | 69.26% | 69.53% | 68.80% | 67.54% | **65.41%** |
| 0.9 | 100 | | 73.90% | 73.45% | **72.31%** | 68.19% | 55.91% |
| 0.9 | 256 | | **73.98%** | **73.84%** | 72.25% | **69.50%** | 62.52% |
| 0.0 | 100 | | 72.72% | 72.83% | 72.90% | **71.93%** | 64.43% |
| 0.0 | 256 | ✓ | 69.57% | 70.14% | 69.06% | 67.90% | **65.27%** |
| 0.9 | 100 | | **74.05%** | 73.62% | 72.84% | 68.83% | 53.42% |
| 0.9 | 256 | | 73.99% | **74.05%** | **73.23%** | 70.50% | 64.63% |

In Figures 13 to 16 we plot the evolution of Lottery Ticket validation accuracy at all studied sparsities. These graphs serve to complement the results from Table 2 in the main paper, and Tables 11 and 12 above. In those tables we highlight accuracies at specific sparsities, which are indicated on the x-axis in the figures. We can see more clearly in these graphs that the more unstable approaches suffer from more significant accuracy degradation throughout the pruning process, even though the dense networks attained with those procedures have superior performances. For the results found with VGG-16 we notice that the inflection point where the more stable hyperparameters start to outperform unstable hyperparameters occurs at a higher sparsity than for the ResNet models. By analysing the error barrier results from Figure 5, as well as the optimal rewind points from Table 15, we notice that VGG-16 is significantly more unstable to SGD noise than the ResNet models studied. Likely this is a result of the different overparameterization of VGG-16 vs ResNet-18, where VGG-16 has ± 138 million parameters, while ResNet-18 only has ± 11.7 million parameters.

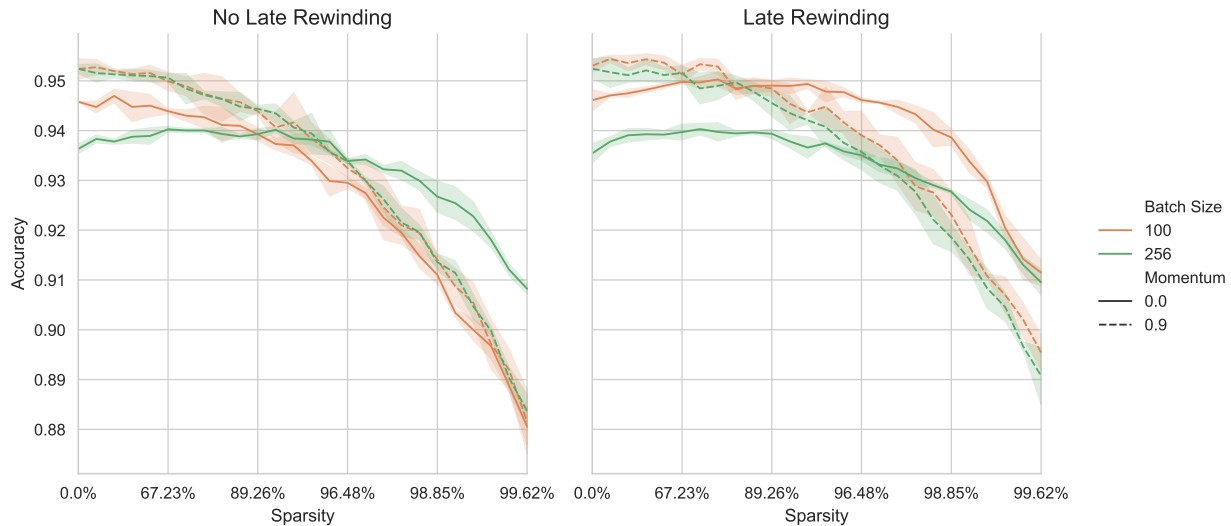

Figure 13: Validation accuracies at different sparsities for Lottery Tickets extracted from a **ResNet-18** model on the **CIFAR-10** dataset.

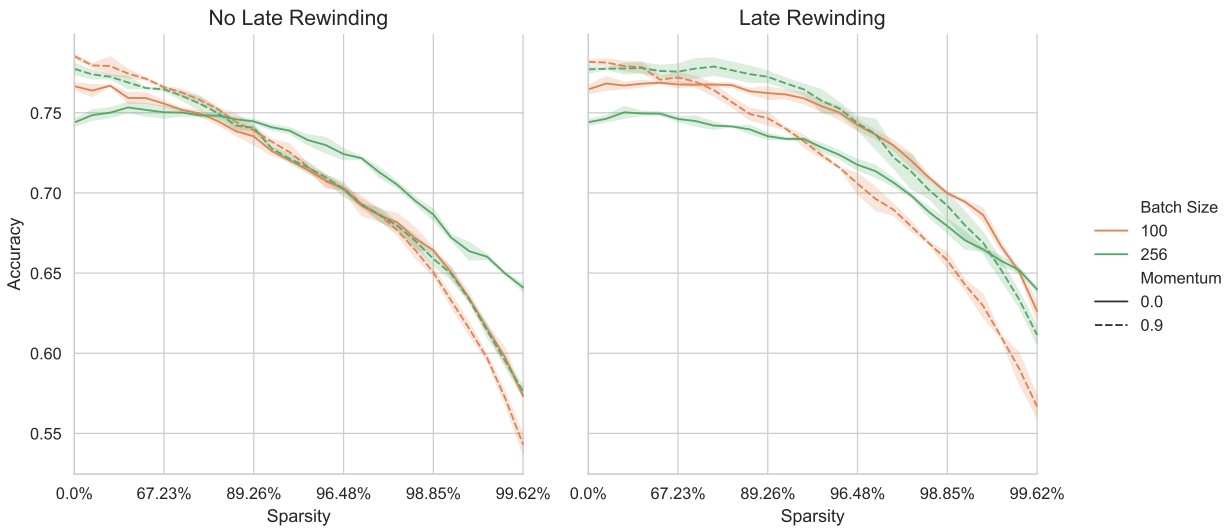

Figure 14: Validation accuracies at different sparsities for Lottery Tickets extracted from a **ResNet-18** model on the **CIFAR-100** dataset.

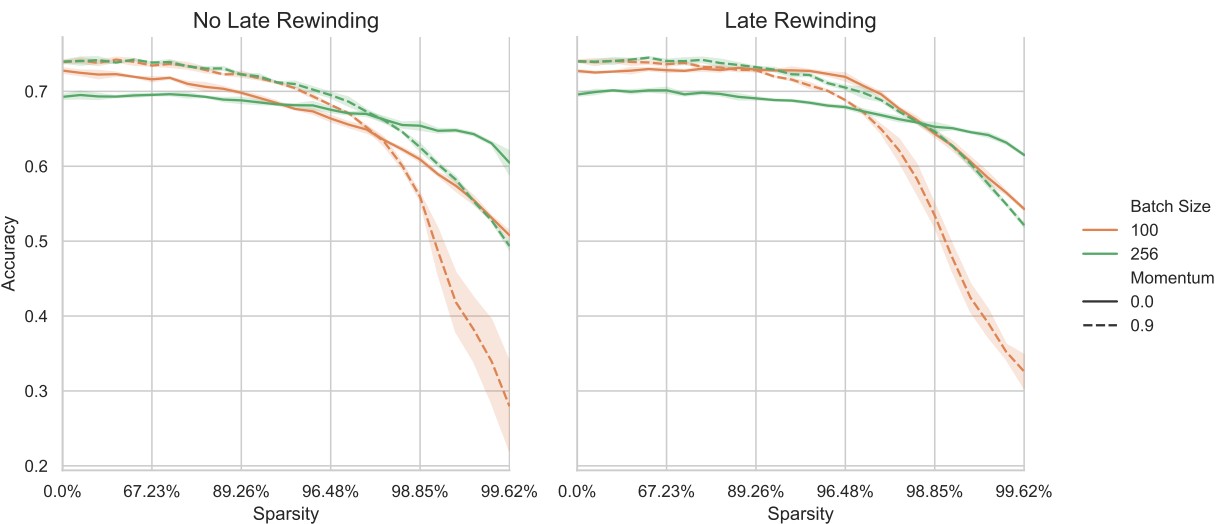

Figure 15: Validation accuracies at different sparsities for Lottery Tickets extracted from a **VGG-16** model on the **CIFAR-100 dataset**.

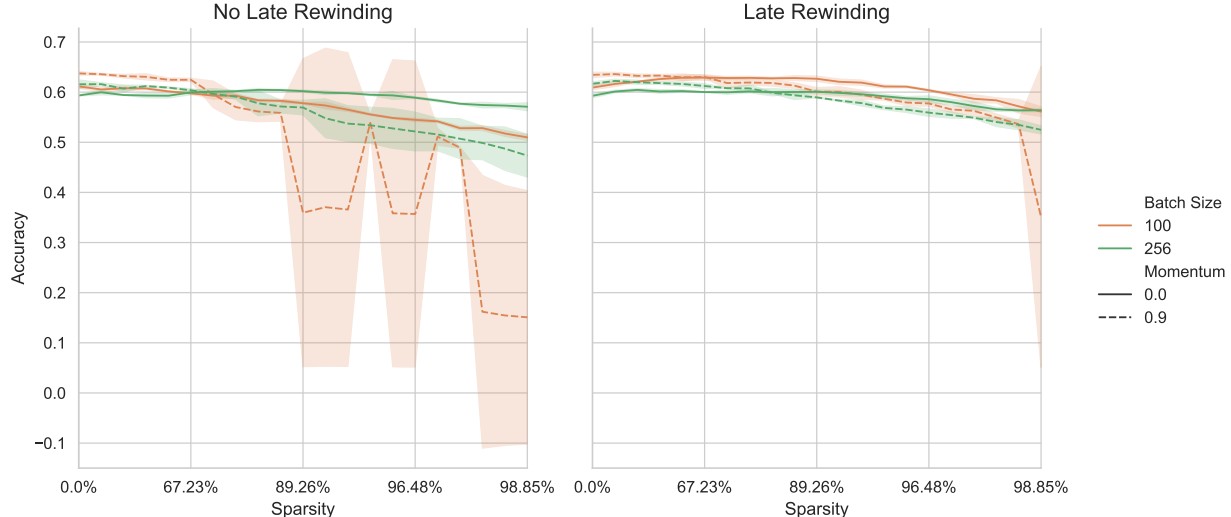

Figure 16: Validation accuracies at different sparsities for Lottery Tickets extracted from a **ResNet-34** model on the **TinyImageNet** dataset. Notice that some iterations record a significant accuracy drop, together with a high standard deviation. This indicates that at that iteration one (or more) random runs did not train and instead remained at random chance (0.5%).

### C.1 More extreme batch sizes

In the main paper, we have demonstrated that lower batch sizes result in more SGD instability, but better generalization of the dense network. Conversely, for higher batch sizes, we see a reduction in SGD instability, which allows for winning Lottery Tickets to be found without pretraining.

In Table 13, we cover a wider variety of batch sizes for ResNet-18 + CIFAR-10 than in Appendix C, including batch sizes which are typically not used in practice. More specifically, this means using very low batch sizes, such as [8, 32] or very high batch sizes such as [1024, 2048]. We show that smaller batch sizes suffer from significant degradation at higher sparsities, but have higher accuracy at the earlier sparsities. Interestingly, we observe cases where the accuracy might suffer from too much stability. For example, comparing $\mu = 0.9$, BS=2048 in the settings with and without late rewinding, we see that there is a non-trivial accuracy drop at higher sparsities in the setting with late rewinding compared to that without late rewinding.

In Figure 17 we clearly see the effect of batch size on instability. We notice that at very large batch sizes, the application of momentum in combination with late rewinding still results in almost stable configurations.

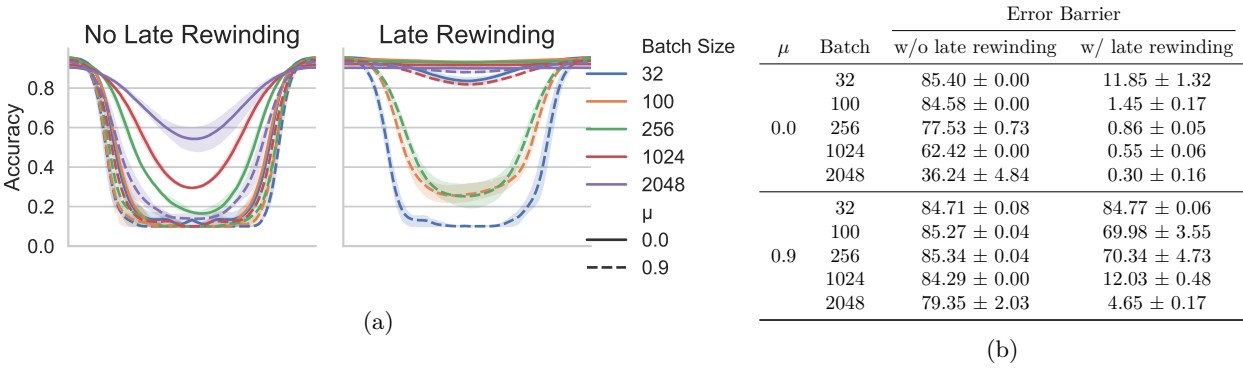

|  |  | | Error Barrier | |
| --- | --- | --- | --- | --- |
| $\mu$ | Batch | | w/o late rewinding | w/ late rewinding |
| | 32 | | $85.40 \pm 0.00$ | $11.85 \pm 1.32$ |
| | 100 | | $84.58 \pm 0.00$ | $1.45 \pm 0.17$ |
| 0.0 | 256 | | $77.53 \pm 0.73$ | $0.86 \pm 0.05$ |
| | 1024 | | $62.42 \pm 0.00$ | $0.55 \pm 0.06$ |
| | 2048 | | $36.24 \pm 4.84$ | $0.30 \pm 0.16$ |
| | 32 | | $84.71 \pm 0.08$ | $84.77 \pm 0.06$ |
| | 100 | | $85.27 \pm 0.04$ | $69.98 \pm 3.55$ |
| 0.9 | 256 | | $85.34 \pm 0.04$ | $70.34 \pm 4.73$ |
| | 1024 | | $84.29 \pm 0.00$ | $12.03 \pm 0.48$ |
| | 2048 | | $79.35 \pm 2.03$ | $4.65 \pm 0.17$ |

(a)

(b)

Figure 17: Instability for a dense **ResNet-18** model trained on **CIFAR10** with different training configurations, featuring more batch sizes. We show the full interpolation curve in **(a)** and the error barrier in **(b)**.

Table 13: Validation accuracy of **ResNet-18** lottery tickets extracted for **CIFAR-10** given more extreme batch sizes at different sparsity levels. CAUTION: these results were attained with only a single run each. Winning tickets are underlined.

| Ticket Parameters | | | Sparsity | | | | |
|---|---|---|---|---|---|---|---|
| $\mu$ | Pretrain | Batch Size | 0.00% | 67.23% | 89.26% | 96.48% | 98.85% |
| 0.0 | ✗ | 8 | $95.29 \pm 0.00\%$ | $\underline{\mathbf{95.54 \pm 0.00\%}}$ | $\mathbf{94.81 \pm 0.00\%}$ | $\mathbf{93.79 \pm 0.00\%}$ | $\mathbf{91.93 \pm 0.00\%}$ |
| | | 32 | $\mathbf{95.41 \pm 0.00\%}$ | $95.29 \pm 0.00\%$ | $94.71 \pm 0.00\%$ | $93.68 \pm 0.00\%$ | $91.72 \pm 0.00\%$ |
| | | 1024 | $91.87 \pm 0.00\%$ | $91.81 \pm 0.00\%$ | $91.62 \pm 0.00\%$ | $91.24 \pm 0.00\%$ | $89.86 \pm 0.00\%$ |
| | | 2048 | $90.92 \pm 0.00\%$ | $90.74 \pm 0.00\%$ | $90.37 \pm 0.00\%$ | $90.00 \pm 0.00\%$ | $88.78 \pm 0.00\%$ |
| 0.9 | ✗ | 8 | $92.92 \pm 0.00\%$ | $88.31 \pm 0.00\%$ | $77.92 \pm 0.00\%$ | $70.27 \pm 0.00\%$ | $10.00 \pm 0.00\%$ |
| | | 32 | $\mathbf{94.53 \pm 0.00\%}$ | $93.75 \pm 0.00\%$ | $91.49 \pm 0.00\%$ | $90.02 \pm 0.00\%$ | $88.62 \pm 0.00\%$ |
| | | 1024 | $94.03 \pm 0.00\%$ | $\underline{\mathbf{94.30 \pm 0.00\%}}$ | $93.54 \pm 0.00\%$ | $92.80 \pm 0.00\%$ | $90.57 \pm 0.00\%$ |
| | | 2048 | $93.14 \pm 0.00\%$ | $\underline{93.46 \pm 0.00\%}$ | $\mathbf{93.60 \pm 0.00\%}$ | $\mathbf{93.54 \pm 0.00\%}$ | $\mathbf{92.31 \pm 0.00\%}$ |
| 0.0 | ✓ | 8 | $95.28 \pm 0.00\%$ | $95.24 \pm 0.00\%$ | $\underline{\mathbf{95.54 \pm 0.00\%}}$ | $\mathbf{94.86 \pm 0.00\%}$ | $93.34 \pm 0.00\%$ |
| | | 32 | $\mathbf{95.56 \pm 0.00\%}$ | $\mathbf{95.39 \pm 0.00\%}$ | $95.52 \pm 0.00\%$ | $94.78 \pm 0.00\%$ | $\mathbf{93.38 \pm 0.00\%}$ |
| | | 1024 | $92.12 \pm 0.00\%$ | $91.92 \pm 0.00\%$ | $91.70 \pm 0.00\%$ | $91.60 \pm 0.00\%$ | $89.87 \pm 0.00\%$ |
| | | 2048 | $90.19 \pm 0.00\%$ | $\underline{90.27 \pm 0.00\%}$ | $\underline{90.33 \pm 0.00\%}$ | $\underline{90.29 \pm 0.00\%}$ | $88.17 \pm 0.00\%$ |
| 0.9 | ✓ | 8 | $92.31 \pm 0.00\%$ | $90.93 \pm 0.00\%$ | $88.98 \pm 0.00\%$ | $87.20 \pm 0.00\%$ | $63.56 \pm 0.00\%$ |
| | | 32 | $\mathbf{94.57 \pm 0.00\%}$ | $93.74 \pm 0.00\%$ | $93.08 \pm 0.00\%$ | $91.63 \pm 0.00\%$ | $89.78 \pm 0.00\%$ |
| | | 1024 | $94.22 \pm 0.00\%$ | $\mathbf{93.89 \pm 0.00\%}$ | $\mathbf{93.96 \pm 0.00\%}$ | $\mathbf{93.22 \pm 0.00\%}$ | $\mathbf{91.69 \pm 0.00\%}$ |
| | | 2048 | $92.90 \pm 0.00\%$ | $\underline{92.92 \pm 0.00\%}$ | $\underline{93.05 \pm 0.00\%}$ | $92.47 \pm 0.00\%$ | $90.91 \pm 0.00\%$ |

## C.2 Additional values for $\mu$

While it is most common to use $\mu = 0.9$ or $\mu = 0.0$ in training configurations for neural networks, we will explore additional values for this momentum parameter in this section. With this we can provide a more granular overview of the effects on the validation accuracy of the resulting lottery tickets. By comparing the granular momentum results from Table 14 with those found using the common values of $\mu$ in Table 11, we notice that our observations in the main paper also holds true for the more granular values of $\mu$. Indeed, solutions found with a lower value of $\mu$ exhibit lower validation accuracy for the dense network, but have a less severe accuracy drop than those found with higher values of $\mu$. Interestingly the results with $\mu = 0.3$ slightly outperform the results with $\mu = 0.0$, indicating that some small amount of momentum can be beneficial.

Table 14: Validation accuracy of **ResNet-18** lottery tickets extracted for **CIFAR-10** with different values for $\mu$ at different sparsity levels. Winning tickets are underlined.

| Ticket Parameters | | | Sparsity | | | | |
|---|---|---|---|---|---|---|---|
| Batch Size | Pretrain | $\mu$ | 0.00% | 67.23% | 89.26% | 96.48% | 98.85% |
| 256 | ✗ | 0.3 | $93.95 \pm 0.36\%$ | $93.92 \pm 0.08\%$ | $\underline{\mathbf{94.07 \pm 0.23\%}}$ | $\mathbf{93.92 \pm 0.27\%}$ | $\mathbf{92.96 \pm 0.17\%}$ |
| | | 0.6 | $94.49 \pm 0.23\%$ | $\mathbf{94.31 \pm 0.16\%}$ | $93.76 \pm 0.30\%$ | $92.82 \pm 0.14\%$ | $90.86 \pm 0.17\%$ |
| | | 0.95 | $\mathbf{95.15 \pm 0.11\%}$ | $94.29 \pm 0.06\%$ | $93.21 \pm 0.89\%$ | $91.58 \pm 1.12\%$ | $89.89 \pm 1.34\%$ |
| 256 | ✓ | 0.3 | $93.97 \pm 0.12\%$ | $\underline{94.24 \pm 0.09\%}$ | $\underline{94.24 \pm 0.11\%}$ | $\underline{94.04 \pm 0.17\%}$ | $93.30 \pm 0.34\%$ |
| | | 0.6 | $94.24 \pm 0.13\%$ | $\underline{\mathbf{94.75 \pm 0.17\%}}$ | $\underline{\mathbf{94.72 \pm 0.05\%}}$ | $\underline{\mathbf{94.60 \pm 0.26\%}}$ | $\mathbf{93.51 \pm 0.28\%}$ |
| | | 0.95 | $\mathbf{95.08 \pm 0.08\%}$ | $94.70 \pm 0.11\%$ | $94.19 \pm 0.31\%$ | $92.90 \pm 0.08\%$ | $91.18 \pm 0.19\%$ |

## C.3 Finding optimal rewind points

For several parameter configurations of ResNet-18 + CIFAR-10, the default rewind point of 2 epochs was insufficient to provide stability to SGD noise. As such, we conduct a study to determine at which points (measured in epochs) SGD stability emerges. We follow the same definition of stability as in Frankle et al. (2020), meaning if the error across the linear path is less than 2%, the weights are considered stable to SGD noise. Note that this definition assumes an absolute difference of 2%, not relative difference to the accuracies.

To speed up this procedure, we use a binary search algorithm to determine the first stable rewind point for each configuration. For this algorithm to work, we assume that stability to SGD noise increases monotonic, which we have validated experimentally. This allows us to find this rewind point in $\lceil log_2 200 \rceil + 1 = 9$ steps. We employ three random initializations for the rewind point, and to measure stability we use three different sets of SGD noise, for a total of 9 runs per configuration. In total this results in $\sim 81$ full training runs for each configuration.

We list the results in Table 15. We notice that, as discussed in the main paper, both momentum and lower batch size impact stability negatively. In particular, the use of momentum has a significant impact, where we notice that in those cases stability only emerges at the earliest at around 10% of the total training epochs. In the worst case scenario, stability is only achieved at around 42% of the total training time. Also notice that VGG-16 is significantly more unstable than the ResNet models. This is likely due to different levels of overparameterization, which results in much more degrees of freedom of the loss landscape.

Table 15: The first epoch at which stability to SGD noise occurs for the considered model and dataset combinations trained under different hyperparameter settings.

| Network | Dataset | $\mu = 0.0$ | | $\mu = 0.9$ | |
|---|---|---|---|---|---|
| | | 100 BS | 256 BS | 100 BS | 256 BS |
| ResNet-18 | CIFAR-10 | $2.3 \pm 0.5$ | $1.7 \pm 0.5$ | $42.7 \pm 1.9$ | $23.7 \pm 1.2$ |
| ResNet-18 | CIFAR-100 | $3.0 \pm 0.0$ | $2.0 \pm 0.0$ | $47.3 \pm 2.1$ | $19.0 \pm 0.8$ |
| VGG16 | CIFAR-100 | $13.3 \pm 0.5$ | $8.3 \pm 0.5$ | $85.0 \pm 0.8$ | $40.3 \pm 0.5$ |
| ResNet-34 | TinyImageNet | $5.0 \pm 0.0$ | $3.7 \pm 0.9$ | $36.7 \pm 1.9$ | $18.0 \pm 1.4$ |

## D  Additional results for limited Mask Search Budgets

Comparing the AIMP results in Table 16 with the IMP results in Table 11, we notice that tickets found with $\mu = 0.9$, BS = 100 at 20eps outperform the baseline IMP results, while at 50eps they underperform those results. The more stable approaches with $\mu = 0.0$ have worse performance for both AIMP budgets, while the approach with $\mu = 0.9$, BS=256 has a roughly similar performance between AIMP and IMP.

Table 16: AIMP results for different hyperparameter settings of **ResNet-18** on **CIFAR-10**.

| Budget | $\mu$ | Batch Size | Pretrained | 67.23% | 89.26% | 96.48% | 98.85% |
|---|---|---|---|---|---|---|---|
| 20eps | 0.0 | 100 | ✗ | 94.34% | 93.76% | 92.65% | 90.26% |
| | 0.0 | 256 | | 93.30% | 92.69% | 91.57% | 88.34% |
| | 0.9 | 100 | | **95.18%** | **94.82%** | **93.31%** | **91.75%** |
| | 0.9 | 256 | | 95.01% | 94.55% | 93.25% | 91.18% |
| 50eps | 0.0 | 100 | ✗ | 94.31% | 93.66% | 92.57% | 90.70% |
| | 0.0 | 256 | | 93.71% | 93.38% | 93.08% | 91.50% |
| | 0.9 | 100 | | 95.07% | 94.24% | 92.74% | 90.83% |
| | 0.9 | 256 | | **95.14%** | **94.69%** | **93.65%** | **91.90%** |

## E  Additional feature emergence results

### E.1  Linear Probing

In Figures 18 to 20 we visualize the probes at each sparsity to complement the tabular results of Table 5 in the main paper and Tables 17 and 18 below. We can see that the selection of useful features evolves throughout

the pruning process. While the features are all similar in the randomly initialized dense network, during the first few pruning iterations the most stable approaches show a steady increase in feature performance, while no evolution is noticeable in the most unstable approaches. Furthermore, we can see that those more stable approaches also have higher quality features in the deeper layers, which is not the case for those that are more unstable.

Table 17: Linear probing results at different locations in a **ResNet-18** ticket extracted for **CIFAR-10** under different training configurations. Note that unlike the ResNet-34 + TinyImageNet results in the main paper, we can achieve a good feature extractor without late rewinding, namely for $\mu = 0.0$, batch size = 256, which coincidentally has better instability than the other configurations tested, thus reinforcing our belief.

| Ticket parameters | | | 89.26% sparsity | | 96.48% sparsity | |
|---|---|---|---|---|---|---|
| $\mu$ | Batch Size | Pretrained | Block 4 | Block 8 | Block 4 | Block 8 |
| 0.0 | 100 | | $33.61 \pm 1.60\%$ | $30.93 \pm 0.82\%$ | $33.45 \pm 0.41\%$ | $30.51 \pm 0.38\%$ |
| 0.0 | 256 | ✗ | $\mathbf{51.38 \pm 0.81\%}$ | $\mathbf{82.87 \pm 0.84\%}$ | $\mathbf{52.17 \pm 1.22\%}$ | $\mathbf{80.85 \pm 1.57\%}$ |
| 0.9 | 100 | | $29.48 \pm 2.03\%$ | $28.65 \pm 0.92\%$ | $29.39 \pm 1.81\%$ | $25.71 \pm 1.03\%$ |
| 0.9 | 256 | | $30.70 \pm 0.52\%$ | $27.83 \pm 1.55\%$ | $29.87 \pm 1.42\%$ | $28.15 \pm 0.21\%$ |
| 0.0 | 100 | | $52.08 \pm 0.95\%$ | $73.11 \pm 1.16\%$ | $49.32 \pm 1.95\%$ | $65.81 \pm 1.77\%$ |
| 0.0 | 256 | ✓ | $\mathbf{56.27 \pm 0.48\%}$ | $\mathbf{84.53 \pm 1.07\%}$ | $\mathbf{55.91 \pm 0.69\%}$ | $\mathbf{83.71 \pm 1.35\%}$ |
| 0.9 | 100 | | $39.27 \pm 1.58\%$ | $48.92 \pm 3.44\%$ | $36.74 \pm 0.67\%$ | $42.49 \pm 2.03\%$ |
| 0.9 | 256 | | $37.27 \pm 5.31\%$ | $46.29 \pm 7.13\%$ | $33.97 \pm 1.76\%$ | $38.62 \pm 3.30\%$ |
| Permuted ($\mu = 0.0$, BS=256) | | | $32.57 \pm 1.68\%$ | $33.69 \pm 0.56\%$ | $29.88 \pm 1.36\%$ | $31.81 \pm 0.61\%$ |

Table 18: Linear probing results at different locations in a **ResNet-18** ticket extracted for **CIFAR-100** under different training configurations. Note that unlike the ResNet-34 + TinyImageNet results in the main paper, we can achieve a good feature extractor without late rewinding, namely for $\mu = 0.0$, batch size = 256, which coincidentally has less instability than the other configurations tested, thus reinforcing our belief.

| Ticket parameters | | | 89.26% sparsity | | 96.48% sparsity | |
|---|---|---|---|---|---|---|
| $\mu$ | Batch Size | Pretrained | Block 4 | Block 8 | Block 4 | Block 8 |
| 0.0 | 100 | | $9.50 \pm 0.21\%$ | $9.31 \pm 0.76\%$ | $9.82 \pm 0.45\%$ | $10.51 \pm 0.32\%$ |
| 0.0 | 256 | ✗ | $\mathbf{15.19 \pm 0.41\%}$ | $\mathbf{44.26 \pm 0.88\%}$ | $\mathbf{15.48 \pm 0.35\%}$ | $\mathbf{36.57 \pm 0.87\%}$ |
| 0.9 | 100 | | $7.13 \pm 0.87\%$ | $7.14 \pm 0.27\%$ | $7.29 \pm 0.94\%$ | $7.41 \pm 1.54\%$ |
| 0.9 | 256 | | $8.22 \pm 0.09\%$ | $7.84 \pm 0.08\%$ | $8.73 \pm 0.43\%$ | $8.38 \pm 0.62\%$ |
| 0.0 | 100 | | $16.02 \pm 0.21\%$ | $34.57 \pm 0.48\%$ | $14.32 \pm 0.75\%$ | $22.01 \pm 1.17\%$ |
| 0.0 | 256 | ✓ | $\mathbf{18.40 \pm 0.59\%}$ | $\mathbf{49.56 \pm 0.85\%}$ | $\mathbf{17.52 \pm 0.66\%}$ | $\mathbf{43.71 \pm 1.19\%}$ |
| 0.9 | 100 | | $8.63 \pm 0.16\%$ | $14.37 \pm 1.17\%$ | $8.43 \pm 0.61\%$ | $11.11 \pm 0.28\%$ |
| 0.9 | 256 | | $11.54 \pm 1.21\%$ | $21.07 \pm 2.74\%$ | $11.01 \pm 1.30\%$ | $16.18 \pm 2.03\%$ |
| Permuted ($\mu = 0.0$, BS=256) | | | $8.43 \pm 0.03\%$ | $10.13 \pm 0.61\%$ | $7.18 \pm 0.34\%$ | $4.88 \pm 0.18\%$ |

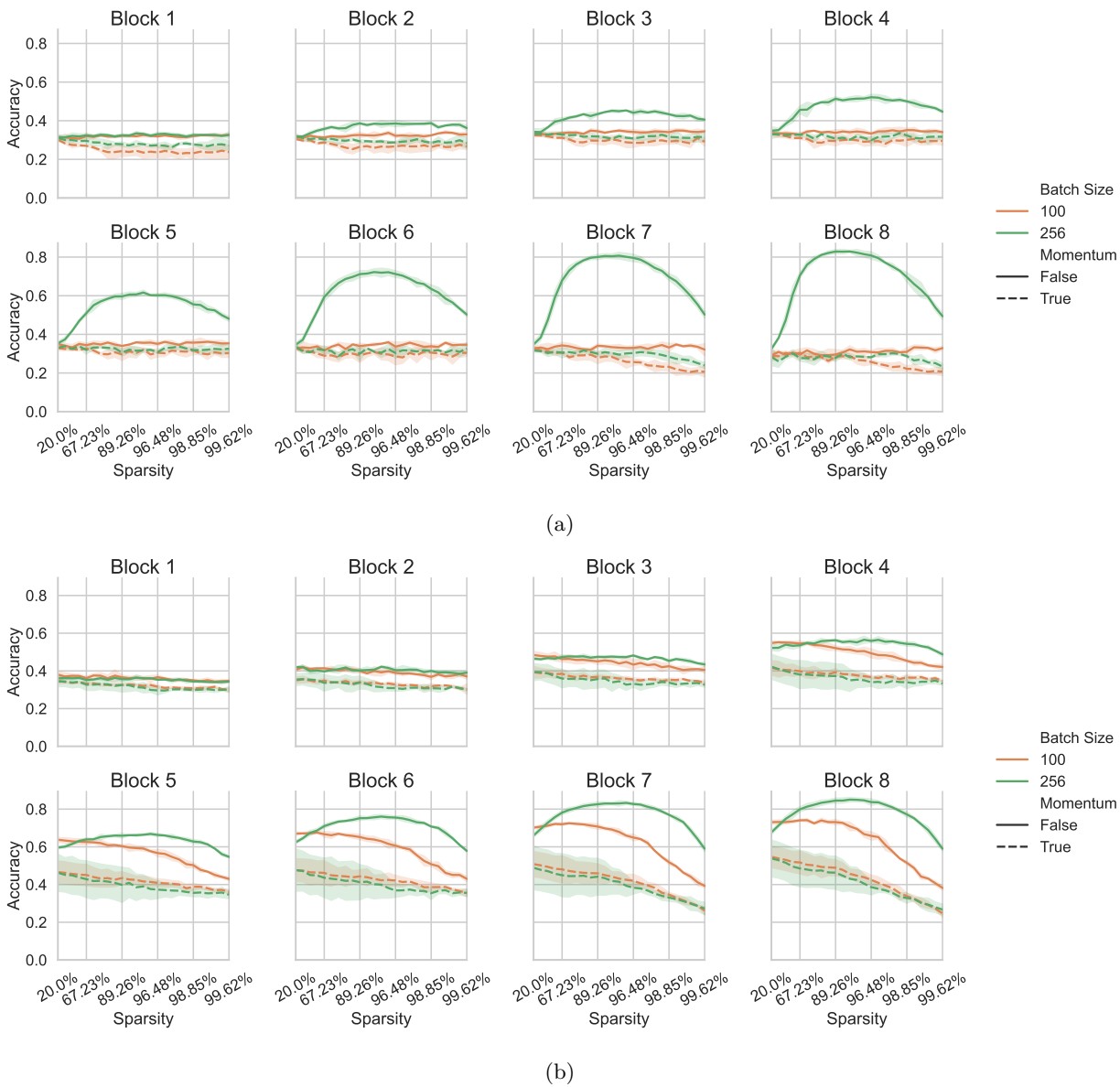

Figure 18: Linear probing results at all sparsities for **ResNet-18** Lottery Tickets on **CIFAR-10** given different hyperparameter configurations. **(a)** Without late rewinding, and **(b)** with late rewinding.

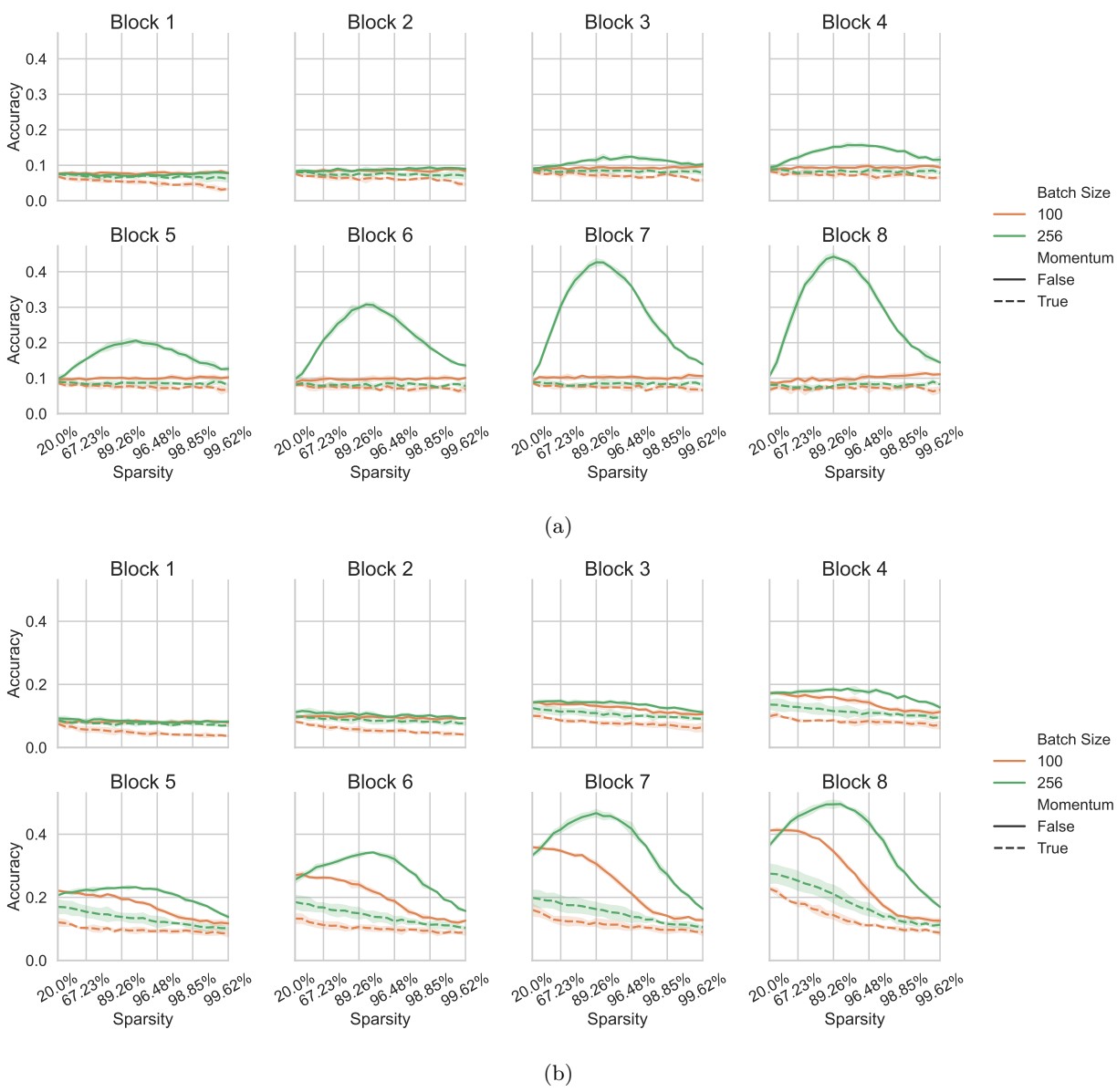

Figure 19: Linear probing results at all sparsities for **ResNet-18** Lottery Tickets on **CIFAR-100** given different hyperparameter configurations. **(a)** Without late rewinding, and **(b)** with late rewinding.

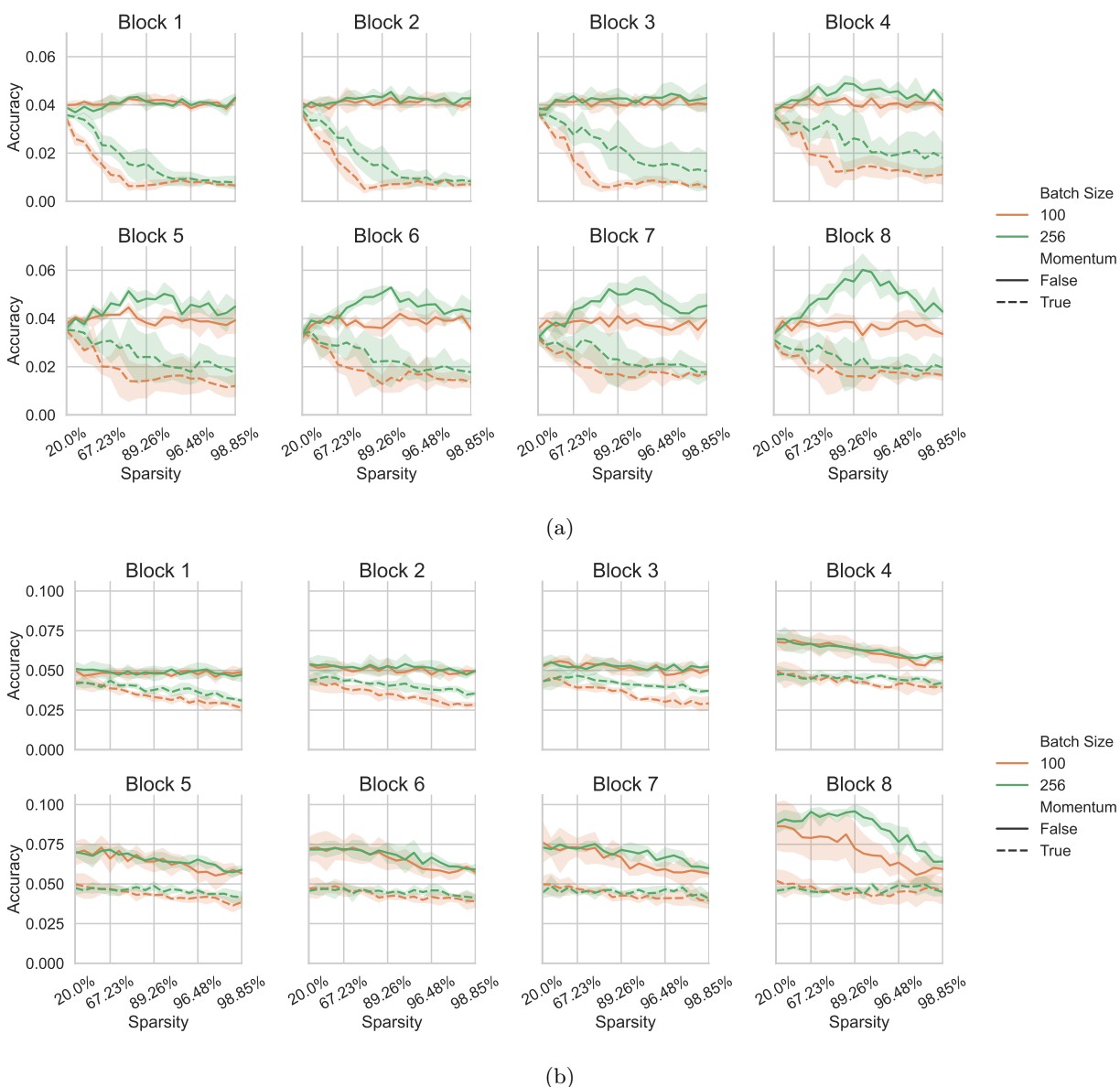

Figure 20: Linear probing results at all sparsities for **ResNet-34** Lottery Tickets on **TinyImageNet** given different hyperparameter configurations. **(a)** Without late rewinding, and **(b)** with late rewinding.

### E.2 Transferability

In Tables 19 and 20 we show the transferability results for frozen tickets extracted for CIFAR-100 and TinyImageNet. When listing the datasets in order of increasing complexity, we can achieve the following list: MNIST < EuroSAT ≈ CIFAR-10 < CIFAR-100 < TinyImageNet. We notice in some experiments that our frozen tickets at initialization don't outperform any of the frozen fully trained dense networks. More specifically, in most of the cases where we transfer from a more complex dataset to a less complex dataset, the trained dense network outperforms the best pruned ticket by some margin. Additionally, transferring from frozen CIFAR-100 tickets seems better than transferring from frozen TinyImageNet tickets, either because the CIFAR-100 tickets have less instability, or due to the network structure.

Table 19: Transferability of frozen 89.26% **ResNet-18** tickets extracted on **CIFAR-100** with different configurations. Starred(*) entries correspond to dense trained networks, meaning the feature extractors have a sparsity of 0.00%.

| Ticket parameters | | | Target dataset | | | |
|---|---|---|---|---|---|---|
| $\mu$ | Batch Size | Pretrained | MNIST | CIFAR-10 | TinyImageNet | EuroSAT |
| 0.0 | 100 | | $84.47 \pm 0.80\%$ | $38.17 \pm 0.23\%$ | $8.74 \pm 0.31\%$ | $73.05 \pm 1.00\%$ |
| 0.0 | 256 | ✗ | $\mathbf{97.18 \pm 0.20\%}$ | $\mathbf{67.60 \pm 0.89\%}$ | $\mathbf{20.59 \pm 0.97\%}$ | $\mathbf{86.59 \pm 0.54\%}$ |
| 0.9 | 100 | | $79.71 \pm 0.79\%$ | $34.95 \pm 1.95\%$ | $6.78 \pm 0.07\%$ | $66.08 \pm 1.99\%$ |
| 0.9 | 256 | | $79.34 \pm 3.08\%$ | $35.84 \pm 0.97\%$ | $7.29 \pm 0.61\%$ | $68.76 \pm 0.70\%$ |
| 0.0 | 100 | | $96.83 \pm 0.07\%$ | $65.44 \pm 1.14\%$ | $22.81 \pm 1.07\%$ | $86.67 \pm 1.10\%$ |
| 0.0 | 256 | ✓ | $\mathbf{97.53 \pm 0.18\%}$ | $\mathbf{70.40 \pm 0.48\%}$ | $\mathbf{24.62 \pm 0.51\%}$ | $\mathbf{89.03 \pm 0.09\%}$ |
| 0.9 | 100 | | $91.92 \pm 0.58\%$ | $47.79 \pm 1.06\%$ | $11.27 \pm 1.61\%$ | $74.14 \pm 1.63\%$ |
| 0.9 | 256 | | $95.47 \pm 0.46\%$ | $53.86 \pm 2.86\%$ | $16.44 \pm 1.08\%$ | $80.78 \pm 1.61\%$ |
| Frozen dense* ($\mu = 0.0$, BS=256) | | | $95.83 \pm 0.24\%$ | $78.12 \pm 0.11\%$ | $17.10 \pm 0.97\%$ | $79.14 \pm 2.16\%$ |
| Frozen dense* ($\mu = 0.9$, BS=100) | | | $90.10 \pm 0.36\%$ | $75.39 \pm 0.41\%$ | $13.16 \pm 0.46\%$ | $67.04 \pm 3.70\%$ |

Table 20: Transferability of a frozen 89.26% **ResNet-34** tickets extracted on **TinyImageNet** with different configurations. Starred(*) entries correspond to dense trained networks, meaning the feature extractors have a sparsity of 0.00%.

| Ticket parameters | | | Target dataset | | | |
|---|---|---|---|---|---|---|
| $\mu$ | Batch Size | Pretrained | MNIST | CIFAR-10 | CIFAR-100 | EuroSAT |
| 0.0 | 100 | | $66.05 \pm 1.23\%$ | $27.87 \pm 0.60\%$ | $10.25 \pm 0.63\%$ | $63.25 \pm 0.76\%$ |
| 0.0 | 256 | ✗ | $\mathbf{92.29 \pm 2.16\%}$ | $\mathbf{46.72 \pm 1.98\%}$ | $\mathbf{22.86 \pm 1.06\%}$ | $\mathbf{78.19 \pm 0.30\%}$ |
| 0.9 | 100 | | $62.50 \pm 11.85\%$ | $26.54 \pm 1.83\%$ | $6.98 \pm 1.12\%$ | $46.19 \pm 8.49\%$ |
| 0.9 | 256 | | $67.25 \pm 4.44\%$ | $26.83 \pm 2.69\%$ | $8.87 \pm 1.71\%$ | $54.25 \pm 5.52\%$ |
| 0.0 | 100 | | $93.38 \pm 1.46\%$ | $50.19 \pm 3.07\%$ | $26.99 \pm 2.08\%$ | $80.38 \pm 1.01\%$ |
| 0.0 | 256 | ✓ | $\mathbf{96.56 \pm 0.45\%}$ | $\mathbf{59.00 \pm 1.22\%}$ | $\mathbf{35.73 \pm 2.02\%}$ | $\mathbf{87.88 \pm 1.04\%}$ |
| 0.9 | 100 | | $85.83 \pm 3.89\%$ | $40.15 \pm 1.69\%$ | $13.60 \pm 1.41\%$ | $68.28 \pm 0.63\%$ |
| 0.9 | 256 | | $86.63 \pm 5.18\%$ | $38.01 \pm 2.34\%$ | $13.23 \pm 1.79\%$ | $68.76 \pm 3.24\%$ |
| Frozen dense* ($\mu = 0.0$, BS=256) | | | $92.77 \pm 0.00\%$ | $62.58 \pm 0.09\%$ | $40.91 \pm 0.46\%$ | $90.10 \pm 0.45\%$ |
| Frozen dense* ($\mu = 0.9$, BS=100) | | | $92.79 \pm 0.56\%$ | $65.11 \pm 0.19\%$ | $44.24 \pm 0.02\%$ | $90.28 \pm 0.52\%$ |

## F  Additional few-shot learning results

In Figures 21 to 23 we visualize the full few-shot results at all sparsities to complement the tabular results recorded in Table 7 in the main paper and Tables 21 and 22 below. As with the linear probing results, we notice that the initial dense network has limited propensity for few-shot learning, but this increases for the more stable approaches during the initial few pruning iterations. This shows that the better few-shot performance is associated with the linear probing efficacy. Indeed, it is quite logical to assume that if good features are already encoded in the network, the network has a 'headstart' for training with limited data, and consequently can achieve better accuracy with limited samples.

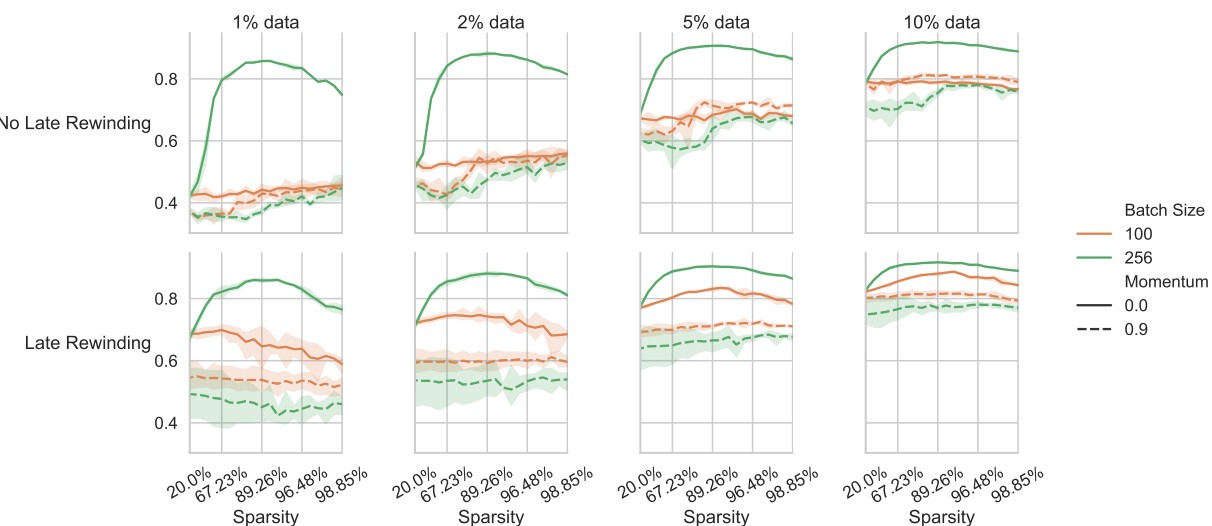

Figure 21: Few-shot performances at all sparsities for **ResNet-18** Lottery Tickets on **CIFAR-10** given different hyperparameter configurations.

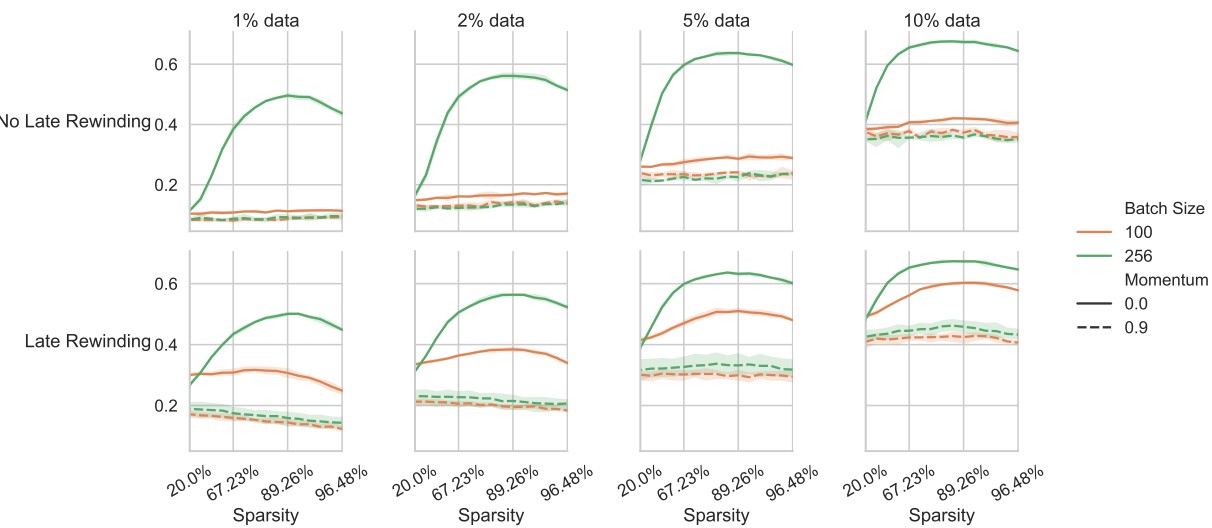

Figure 22: Few-shot performances at all sparsities for **ResNet-18** Lottery Tickets on **CIFAR-100** given different hyperparameter configurations.

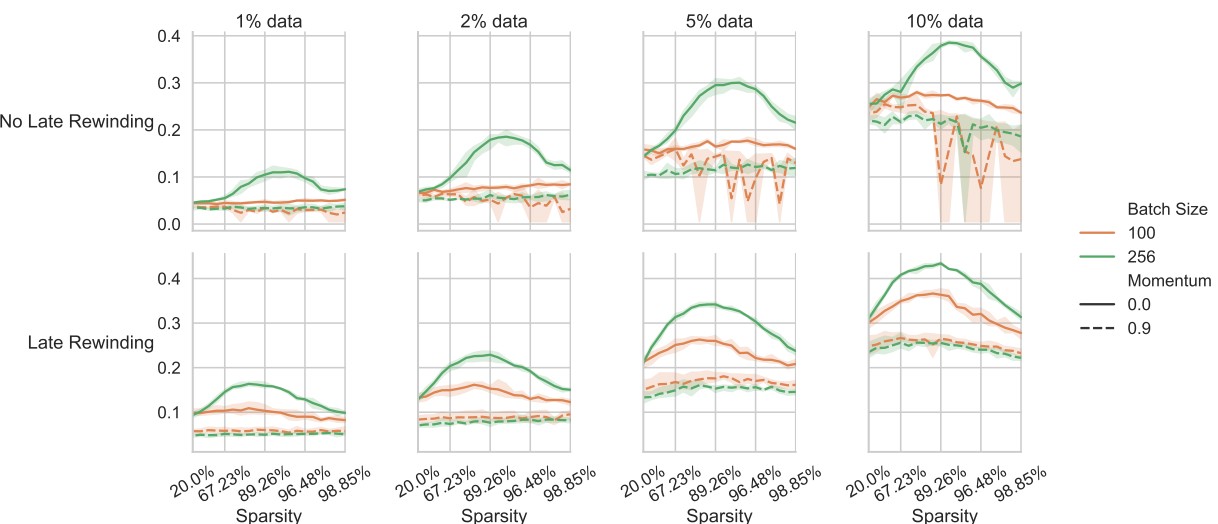

Figure 23: Few-shot performances at all sparsities for **ResNet-34** Lottery Tickets on **TinyImageNet** given different hyperparameter configurations.

Table 21: Validation accuracies of a 89.26% sparse **ResNet-10** ticket when trained on a **CIFAR-10** subset.

| | Ticket parameters | | | Subset sizes | | |
|---|---|---|---|---|---|---|
| $\mu$ | Batch Size | Pretrained | 1% | 2% | 5% | 10% |
| 0.0 | 100 | | $44.22 \pm 1.47\%$ | $53.37 \pm 1.41\%$ | $68.40 \pm 0.96\%$ | $78.90 \pm 0.30\%$ |
| 0.0 | 256 | ✗ | $\mathbf{85.43 \pm 0.58\%}$ | $\mathbf{88.03 \pm 0.62\%}$ | $\mathbf{90.58 \pm 0.21\%}$ | $\mathbf{91.82 \pm 0.11\%}$ |
| 0.9 | 100 | | $42.93 \pm 2.12\%$ | $52.80 \pm 3.05\%$ | $71.35 \pm 1.49\%$ | $80.96 \pm 0.29\%$ |
| 0.9 | 256 | | $37.02 \pm 1.03\%$ | $47.38 \pm 1.35\%$ | $63.93 \pm 4.25\%$ | $75.44 \pm 1.05\%$ |
| 0.0 | 100 | | $68.02 \pm 2.81\%$ | $74.49 \pm 0.60\%$ | $83.43 \pm 0.25\%$ | $88.69 \pm 0.48\%$ |
| 0.0 | 256 | ✓ | $\mathbf{85.84 \pm 0.70\%}$ | $\mathbf{88.09 \pm 0.47\%}$ | $\mathbf{90.40 \pm 0.21\%}$ | $\mathbf{91.68 \pm 0.11\%}$ |
| 0.9 | 100 | | $53.85 \pm 3.47\%$ | $59.29 \pm 2.70\%$ | $71.10 \pm 0.55\%$ | $81.51 \pm 0.90\%$ |
| 0.9 | 256 | | $45.06 \pm 4.52\%$ | $53.55 \pm 5.55\%$ | $66.54 \pm 0.32\%$ | $76.99 \pm 0.75\%$ |
| | *Dense Network* | | $42.54 \pm 1.15\%$ | $51.25 \pm 0.78\%$ | $67.15 \pm 0.34\%$ | $78.42 \pm 0.53\%$ |

Table 22: Validation accuracies of a 89.26% sparse **ResNet-18** ticket when trained on a **CIFAR-100** subset.

| | Ticket parameters | | | Subset sizes | | |
|---|---|---|---|---|---|---|
| $\mu$ | Pretraining | Batch Size | 1% | 2% | 5% | 10% |
| 0.0 | 100 | | $11.20 \pm 0.13\%$ | $16.71 \pm 0.40\%$ | $28.61 \pm 0.40\%$ | $42.01 \pm 0.21\%$ |
| 0.0 | 256 | ✗ | $\mathbf{49.60 \pm 0.53\%}$ | $\mathbf{56.13 \pm 0.84\%}$ | $\mathbf{63.70 \pm 0.26\%}$ | $\mathbf{67.36 \pm 0.44\%}$ |
| 0.9 | 100 | | $8.63 \pm 0.31\%$ | $14.23 \pm 0.42\%$ | $24.13 \pm 0.28\%$ | $37.31 \pm 0.78\%$ |
| 0.9 | 256 | | $9.24 \pm 0.68\%$ | $13.43 \pm 0.61\%$ | $22.54 \pm 1.03\%$ | $35.58 \pm 0.49\%$ |
| 0.0 | 100 | | $30.69 \pm 1.38\%$ | $38.44 \pm 0.70\%$ | $51.03 \pm 0.72\%$ | $60.29 \pm 0.18\%$ |
| 0.0 | 256 | ✓ | $\mathbf{50.10 \pm 0.39\%}$ | $\mathbf{56.36 \pm 0.31\%}$ | $\mathbf{63.23 \pm 0.50\%}$ | $\mathbf{67.32 \pm 0.35\%}$ |
| 0.9 | 100 | | $14.44 \pm 0.97\%$ | $19.51 \pm 0.61\%$ | $29.91 \pm 1.60\%$ | $42.48 \pm 1.64\%$ |
| 0.9 | 256 | | $15.91 \pm 2.34\%$ | $21.50 \pm 2.43\%$ | $33.17 \pm 3.51\%$ | $45.86 \pm 1.76\%$ |
| | *Dense Network* | | $9.44 \pm 0.48\%$ | $13.69 \pm 0.45\%$ | $22.73 \pm 0.09\%$ | $34.34 \pm 0.48\%$ |

### F.1 Using forgetting scores as selection criterion

The results earlier in this section, and in Section 5.2 in the main paper are achieved with randomly selected class-balanced subsets from the training dataset. In this experiment we study a more robust selection criterion by using sample-wise forgetting statistics from Toneva et al. (2019) to generate class-balanced dataset subsets. We show the results for CIFAR-10 subsets on the 89.26% sparse ticket achieved with $\mu = 0.0$, batch size 256 in Table 23. Compared to the random selection results in Table 21, we notice that random selection is indeed a robust baseline at this data sparsity, as forgetting statistics fail to convincingly outperform. We however notice that interestingly for ticket the 10% most forgotten samples generalize better, while for the permuted ticket and dense network the least-forgotten samples perform better at 10% subset size.

Table 23: Validation accuracies for a 89.26% sparse ticket, a permuted version of the ticket and a dense **ResNet-18** model trained on different dataset subsets of **CIFAR-10**. Each model was achieved with $\mu = 0.0$, batch size 256.

| Network | Least forgotten subset | | Most forgotten subset | |
|---|---|---|---|---|
| | 1% | 10% | 1% | 10% |
| Ticket | $85.53 \pm 0.12\%$ | $87.28 \pm 0.04\%$ | $82.32 \pm 0.91\%$ | $91.40 \pm 0.15\%$ |
| Permuted | $47.12 \pm 0.80\%$ | $68.62 \pm 0.25\%$ | $31.51 \pm 0.50\%$ | $47.21 \pm 0.26\%$ |
| Dense | $46.54 \pm 0.68\%$ | $68.21 \pm 0.52\%$ | $31.87 \pm 0.38\%$ | $45.52 \pm 0.74\%$ |

## G Hyperparameter configurations

**General parameters.** Each training run contains of 200 epochs with a chosen batch size and momentum parameter. Throughout training the learning rate starts at 0.1 (0.2 for TinyImageNet) and is cosine annealed. We also apply a weight decay of 1E-4. Each dataset uses normalization, random cropping and horizontal flipping as data augmentation.

**LTH parameters.** Each iteration 20% of the unpruned parameters are pruned. Late-rewinding involves training for 2 epochs to determine the rewinding point. Each experiments consists of 25 (20 for TinyImageNet) rounds of IMP.

**Linear Probing.** Each linear probe is trained for 5 epochs with a learning rate of 0.005 and weight decay of 1E-4. No LR scheduling is applied in this case.

