# OpenReview forum: "The Influence of SGD noise on Lottery Ticket Performance"
_TMLR — Rejected by TMLR_

### Review · Reviewer_6jnC · 2025-02-11

**Summary Of Contributions:**

The paper focuses on SGD instability caused by lottery ticket methods, and proposes to use improved hyperparameters to mitigate this problem. The authors focus on two hyperparameters, batch size and momentum, and analyze their effect on SGD instability, i.e., the ability of a sparse network to find the same loss basin in-spite of SGD noise (due to different batch ordering). Using these insights, the authors observe the effects on lottery ticket identification via experiments. The main conclusion drawn by the authors is that SGD instability is not always indicative of the LT performance but depends on the hyperparameters used.

**Audience:**

Yes

**Broader Impact Concerns:**

No ethical concerns.

**Claims And Evidence:**

No

**Requested Changes:**

1. Since the authors only focus on two hyperparameters, batch size and momentum, it would be important to understand the effects of these in detail. The current experiments only use two values for each hyperparameter to draw rather strong conclusions about their effects. The absence of momentum could also significantly harm generalization performance, but this is unclear since only one dataset is used.
2. The forgetting analysis can be improved to add a clear connection to SGD instability and its relevance to LTs, along with a discussion of how it would affect generalization which is missing at the moment.
3. The authors should also discuss the other factors that can contribute to SGD noise and potentially why batch size and momentum are most important.
4. The discussion on how lack of momentum can affect generalization is missing and this could lead to overall poor optimization of LTs.

While the authors address an important and interesting topic, I believe they require some revisions to their analysis before I can recommend acceptance.

**Strengths And Weaknesses:**

1. The authors identify that no momentum and high batch sizes reduces instability. Experiments show that SGD noise can be controlled by momentum and batch size as examples and is not entirely indicative of the ability to find LTs.
2. Without momentum convergence is also faster, but the performance could be much worse. If the overall performance is worse, even if the lottery ticket training succeeds, if it cannot achieve the higher performance (with momentum), can it be deemed a lottery ticket? Since this subnetwork still struggles to train.
3. The forgetting analysis must be clearly linked with SGD noise, otherwise it is unclear what the objective of this analysis is. Does SGD noise enhance forgetting? Table 1 needs an elaborate explanation, including metrics being used.
4. The authors find that differences due to SGD instability are exaggerated in shortened training durations. In Table 3, why is training shorter (20 epochs) better for 95% sparsity than training longer (50 epochs).
5. The linear probing results on Tiny ImageNet to interpret features learnt by the mask at each layer seem inconclusive to me. A random prediction for Tiny ImageNet should be 0.5%, and the linear probe accuracies are very low, not too much better than random. Moreover, the differences between earlier layers and later layers in sometimes non-existent, suggesting that even at later layers the features are not informative, why is this the case? Even in the case of stable tickets, the later layers seem to learn only marginally better features.
6. The authors also observe that transferability is aided by SGD stability.
7. With stable training, smaller dataset sizes can be used to recover performance.

Given the analysis, it is difficult to draw a conclusion based on the results provided.
1. No momentum improves stability but comes at the cost of generalization performance. Is this an acceptable scenario for finding LTs?
2. The discussion also claims that increased training would reduce the error barrier (improve instability) however, this seems unlikely and there are no experiments provided to back this up.
3. Why do the authors only look at a limited number of batch size and momentum configurations, the analysis might benefit from having more intermediate values of momentum to understand its effect on the optimization and the trade-off between generalization and stability.

---

> ### Author Response · Authors · 2025-03-02
> **Reply to reviewer 6jnC (1/2)**
>
> Dear reviewer 6jnC,
>
> We thank you for your review of our manuscript.
>
>
> **Linear Probing.**
> We respectfully disagree with your remark that the linear probing results are inconclusive. As mentioned by you, the random chance accuracy of TinyImageNet is 0.5%. We demonstrate up to 9.5% validation accuracy simply by training a linear classifier on top of the frozen ticket. This represents +-1/6th of the total validation accuracy attained by the classifier when we train the ticket itself. Furthermore, we demonstrate that purely by using the magnitude pruning with reduced instability, we can make a significant gain of > 5% over randomly initialized dense networks. The review also states that the difference between earlier and later layers is minimal and sometimes non-existent. Indeed, for unstable tickets, the difference is non-existent, or even negative in approaches that are unstable, showing that there are no truly useful features encoded in those tickets. For stable tickets however, we respectfully disagree with the observation that the existing differences are minimal. Rather, we see an increase of ~3% when transitioning from the middle layer to the last layer, which represents a significant increase, seeing as the accuracy at block 4 is only around 6%. Furthermore, this increase is not present in the random dense network, nor is it present in the permuted tickets, showing that this is an unique effect of the more stable training, and that useful features are encoded in the ticket. For the other datasets (CIFAR-10, CIFAR-100) which are easier, this effect can be seen even more clearly in App D, Table 14 & 15.
>
> **Only a single dataset.**
> While it is true that in the main paper, we discuss only a single dataset, we have clearly listed in the experimental setup that we study a multitude of different dataset and network combinations, which are listed in the appendix. More specifically, these are also ResNet-18 + CIFAR-10 and ResNet-18 + CIFAR-100. For these datasets, we can make similar observations as for ResNet-34 + TinyImageNet, but we chose to highlight the ResNet-34 results as this combination is the most complex. In the current revision, we additionally include VGG16 + CIFAR-100 to accommodate a request of reviewer gBsZ.
>
> **Limited batch sizes and momentum values.**
> In the appendix, we study more extreme batch sizes for ResNet-18 + CIFAR-10 in B.2. We find that the trends are in general similar to the results attained with the batch sizes used in the main paper. We should note that these results are only attained with a single run, as the runtime for extremely small batch sizes is too long (~10 days) to allow for multiple random runs. The critique on the limited number of momentum values is valid, and as such we have added a new section in the appendix, B.3 where we study three additional different values for the momentum parameter on ResNet-18 + CIFAR-10. Once again, the trends hold for these values, where higher momentum values result in better dense network generalization, but cannot find winning tickets, and have more significant accuracy decrease when pruning.
>
> **On shortened training durations.**
> We discover in Table 3 that certain hyperparameter configurations result in better performing tickets when limiting the budget of Mask Search. This observation only occurs for the more unstable parameters (i.e., mu=0.9, Batch Size = 100), and is not present for more stable parameters (i.e., mu=0.0, Batch Size = 256). Furthermore, we can see even a degradation in accuracy when moving from 20 epochs of budget to 50 epochs of budget for the unstable parameters. This leads us to believe that the reason why IMP for these parameters underperforms is because the accumulation of SGD noise is significantly higher for these parameters, and leads to more difficulty finding the same loss basin. By limiting the accumulation of this noise, we can increase the resulting validation accuracy. Note that for the more stable approach we see a totally different behaviour. As this approach features less SGD noise, we notice that the limited budget (AIMP) has a constant negative effect on the validation accuracy w.r.t. using the full budget (IMP). We have modified Section 4.3 to hopefully make this more clear.

---

> ### Author Response · Authors · 2025-03-02
> **Reply to reviewer 6jnC (2/2)**
>
> **Momentum and generalizability.**
> In this study the main focus is to determine the necessity of the pretraining step for finding winning tickets. This means that we want to see if we can limit instability with other measures than pretraining, and in turn achieve tickets that perform similar or better than the dense network under the same hyperparameter configuration. That means that our goal is not necessarily to find the best possible combinations of pretraining & hyperparameters to find the highest accuracy lottery tickets.
> We have added an additional experiment where we abuse the current approach slightly and decouple the hyperparameters from sparse training and mask search, on the recommendation of reviewer 2Zz7. This allows us to see more clearly whether the performance differences of the tickets are related to the Mask Search phase, or the Sparse Training phase. Here we notice that the hyperparameters which are deemed more stable are the only ones that generate winning tickets irrespective of the way the tickets are trained. Additionally, at higher sparsities, we can see that those still outperform the more unstable tickets under all different training regimes tested for the tickets.
>
> **Forgetting events**
> The forgetting events together with the convergence scores highlight the more exploratory nature of the approaches with momentum. They show that rather than converging to a single loss basin early during training, these approaches go through several loss basins throughout training. We highlight this relationship in Section 6.1. Additionally, in the appendix in section A.1 we make the link between often forgotten samples and instability, showing that the samples which are most often forgotten are the samples that induce the most instability throughout training.
>
> We hope this answers some of your questions and concerns. Please let us know.
>
> Kind regards,
>
> the authors

---

### Review · Reviewer_gBsZ · 2025-02-22

**Summary Of Contributions:**

The paper studies the effect of SGD noise/training instability with respect to different hyperparameters (batch size and momentum) on the final accuracy of the sparse network. The authors use the concept of linear mode connectivity between models trained with different batch orders (but with the same hyper-parameters) to measure the instability for different hyper-parameters. The authors claim that the winning tickets found with HPs stable to SGD noise perform better and are more transferrable. Experiments are conducted on CIFAR-10/100 and Tiny-ImageNet to validate their hypothesis. The paper presents some interesting results (about linear probing in particular). However, experiments do not fully corroborate the claims made in the paper.

**Audience:**

Yes

**Claims And Evidence:**

No

**Requested Changes:**

1. Change the pre-training column with a rewind point (0,5,10, etc.) in Table. 2.
2. Run a sweep over HP for sparse training and use the best HP for training the final sparse network in Table 2.
3. Different rewind point should be evaluated to find the best lottery ticket.
4. In Table 2., add the accuracy of the final pruned model as well.
5. Recent works have shown that IMP solution only works when the LTH and pruned solution lie in the same basin, i.e., there is a LMC between the pruned solutions and trained lottery ticket [1, 2]. The LMC between different iterations of IMP is much a stronger requirement to find lottery tickets and is dependent on SGD noise during training. The authors should at the LMC between different IMP iterations similar to [1], instead of looking at the LMC between two dense models (table in fig1)

### Minor Points:
1. Table in the Fig 1. is added as a figure in the LaTeX.

[1] Paul et al., Unmasking the Lottery Ticket Hypothesis: What's Encoded in a Winning Ticket's Mask?
[2] Utku et al., Gradient Flow in Sparse Neural Networks and How Lottery Tickets Win.

**Strengths And Weaknesses:**

### Strength:
1. The paper tries to study an important problem in sparse training/LTH --- how SGD noise effect the LTH.
2. The results of linear probing are interesting (in Table 4.). Despite lower accuracy of  the dense model with $\mu=0.0$, the linear probing achieves better accuracy for both pretrained and random lottery tickets.

### Weaknesses
1. The main conclusion from the paper is not clear and the experiments do not fully support the claims made in the paper.
2. The author argues that HP that is more stable to SGD noise -- $\mu=0$ --- helps in finding the lottery tickets. However, in Table 2, for moderate sparsities, $\mu=0.9$ performs better for both w/ and w/o weight in rewinding.
3. Authors use the same parameters for finding the mask (using IMP) and for training the final sparse model. Best HP for sparse training should be used to disentangle the effect of HP on finding a good mask and sparse training.
4. In Table 3., $\mu=0.9$ works better for most of the setting, which contradicts the authors' claim that HP more stable to SGD noise find better tickets.
5. There is no information about the rewind point used in the experiments.
6.
> By dampening this instability via smart hyperparameter selection, we show that we can extract ’winning’ tickets without pertaining

I think this is a very strong statement and none of the experiments seem to support this claim. In Tables 2, 3, 4, pretrained lottery tickets perform better.


### Questions:
1. Do you also rewind LR in Algo1. Usually, we rewind the LR schedule as well.
2. In Table 2., what was the rewinding point used?

---

> ### Author Response · Authors · 2025-03-02
> **Reply to reviewer gBsZ (1/2)**
>
> Dear reviewer gBsZ,
>
> We thank you for your thorough review. Below we address your weaknesses and questions sorted by category.
>
> **Clarifying our claims.**
> We do not necessarily claim that hyperparameters which are more stable to SGD noise result in better lottery tickets at all sparsities, but rather we make the following claims (see Abstract): (1) that pretraining is not always necessary to find winning tickets – with winning tickets defined as ‘Sparse subnetworks that outperform the dense network when trained under the same training regime’ [Frankle19], and (2) that at more extreme sparsities the tickets found under less unstable regimes outperform those found under more unstable regimes. While it is true that the ones found with mu = 0.9 have better dense generalization, and accuracy at lower sparsities, it can be seen in Table 1 that none of those tickets are ‘winning’ tickets, as none of them have similar or better accuracy as the dense network under the same hyperparameter configuration. Opposite to that is the case for mu = 0.0 where we see in general lower performance in the earlier sparsities, but we can find winning tickets without pretraining. To make this more clear in Table 2, as well as the related Table 9 in the appendix, we will introduce an additional notation to highlight these, as well as stress the research objective more in the introduction and conclusion. As such, the goal of this study is not to find the best possible combination of hyperparameters that will result in the very best lottery tickets, but rather to explore alternatives for the pretraining phase to limit SGD instability. This in combination with a limited rebuttal period and limited computational resources will not allow us to conduct significant hyperparameter sweeps to fully determine the best possible combination of hyperparameters. That being said, we have added an additional section in the appendix B.2 where we explore different values for mu on ResNet-18 + CIFAR-10. While this may not fully help us to determine the best possible combination, it still provides valuable insights together with appendix B.1 on the influence of these two hyperparameters on the validation accuracy of resulting tickets and the presence or absence of ‘winning’ tickets.
>
> **Using different hyperparameters for sparse training vs mask search.**
> While it is indeed true that the performance of the found lottery tickets could likely be improved by using different hyperparameter settings, this does not follow the original approach of [Frankle19], and would require redefining the definition of ‘winning’ tickets. We do notice the significant impact of some hyperparameters on dense network accuracy and wonder whether this is also applicable to sparse networks. To accommodate this request, and further this understanding, we have conduct a limited grid search on the sparse training hyperparameters using two distinct sets of the ResNet-34 + TinyImageNet tickets. We have added this experiment as an additional subsection (4.4) in the experiments. In brief, we notice that except for the least sparse setting, the more stable ticket outperforms the more unstable ticket no matter the hyperparameters during sparse training. We however want to reiterate that attaining the best possible accuracy is not the main focus of this research.
>
> **On rewinding.**
> You are correct that when rewinding, we also rewind the learning rate schedule back to the rewind point. This follows common practice, but we will make sure that we highlight this more clearly in the experimental setup.
> The information of the rewind point is included in Section 3.3 Experimental Setup ‘In the case of late Rewinding, this budget includes a Pretraining phase of 2 epochs’. This follows the setting of [T22]. As we only use a single rewind point throughout the whole of the paper, this 2 epochs is used in every experiment with late rewinding.
> While additional rewinding points could be considered, these would be expensive experiments to run for ResNet-18 + TinyImageNet (roughly 2000 compute hours per rewind point), which would be infeasible for us in the 2 weeks window. We could provide additional results for ResNet-18 + CIFAR10 however as those experiments are significantly faster. That being said, when extrapolating the results from [Frankle20] regarding the existence of ‘winning’ tickets in relation to linear instability with our experiments in Appendix B.3 that relate to the optimal rewind point for each hyperparameter selection, we would require a significant percentage of the training budget allocated to finding the rewinding point (>10%).

---

> > ### Author Response · Authors · 2025-03-02
> > **Reply to reviewer gBsZ (2/2)**
> >
> > **On Linear Mode Connectivity.**
> > It is true that in some cases where winning tickets exists, the error barrier between two dense solutions is not matching, while it is matching between two sparse solutions, as mentioned in [Paul22]. That being said, in those cases the error barrier is only slightly higher, and not as significantly as in our observations. To accommodate this request, we will endeavor to show results at pruning iteration 1 (i.e., 20% sparsity) in these tables, as well as highlight this phenomenon in the accompanying text. That being said, we do not expect significant differences for the solutions found with mu = 0.9 as we have noticed a requirement for a large number of pretraining epochs for those solutions.
> >
> >
> > We hope that this helps to clarify the goals of this research, and that together with the implemented additional experiments and textual modifications we have improved the quality of the paper.
> >
> > Kind regards,
> > the authors
> >
> > References:
> >
> > [T22] : T et al. 2022 “Sparse Winning Tickets are Data-Efficient Image Recognizers”
> > [Frankle20] : Frankle et al. 2020 “Linear mode connectivity and the lottery ticket hypothesis”
> > [Paul22] : Paul et al. 2022 “Unmasking the Lottery Ticket Hypothesis: What's Encoded in a Winning Ticket's Mask?”

---

> > > ### Author Response · Authors · 2025-03-10
> > > **Addition of LMC between IMP iterations**
> > >
> > > Dear reviewer gBsZ,
> > >
> > > We have added the analysis regarding the linear mode connectivity (error barrier) between subsequent lottery tickets. We discuss this in Section 6.1, as  well as provide results for additional settings in the appendix section B.
> > >
> > > Kind regards,
> > > the authors

---

### Review · Reviewer_2Zz7 · 2025-02-24

**Summary Of Contributions:**

This paper reconsiders the hyperparameter settings required for searching winning tickets in the Lottery Ticket Hypothesis. Previous studies have discussed that the instability of SGD noise is a crucial factor for finding winning tickets, as weight rewinding is necessary for achieving high generalization accuracy; however, this study points out that this factor is insufficient. This paper provides experimental results supporting this claim using multiple metrics and datasets.

**Audience:**

Yes

**Claims And Evidence:**

Yes

**Requested Changes:**

Please see the weakness part.

**Strengths And Weaknesses:**

### Strength
- Related work and background are well-structured.
- The experimental results are well represented through tables and figures.

### Weakness
- The main claim of the paper is unclear. For example, the abstract states that selecting hyperparameters to reduce SGD instability improves the accuracy of winning tickets, while the conclusion argues that SGD instability alone is insufficient to fully explain the accuracy of winning tickets. This part makes me feel that the argument is inconsistent.
- It is unclear which parts of the paper present novel contributions. The Introduction consists entirely of a review of existing studies, making it difficult to understand the motivation, research direction and new contributions of this research.
- The experiments are limited to ResNet, leaving questions about the extent to which the findings generalize to other CNNs, such as VGG, or transformer architectures.
- On page 5, is it a mistake that should be 'μ = 0.0, solid line'?

---

> ### Author Response · Authors · 2025-03-02
>
> Dear reviewer 2Zz7,
>
> Thank you for your review.
>
> **Main Claims of the Paper**
> Our claims are twofold. First we claim that we can find winning tickets without pretraining by smart hyperparameter selection. Winning tickets are defined as following ‘A winning ticket is a sparse subnetwork that can attain similar or better validation accuracy as the dense network under the same training configuration’.
> Our second claim is that even without pretraining we can outperform certain training configurations with pretraining at extreme sparsities.
>
> While we do not explicitly define this ‘extreme’ term – which we resolved in the modified version of the manuscript – we do not claim to outperform all possible configurations at all possible sparsities. However, we can clearly notice in Table 2 that at 89.26% sparsity, the solution with mu = 0.0, 256 batch size, no pretraining already outperforms the majority of the other configurations, as well as being the single winning configuration that does not employ late rewinding. To this end, we have introduced additional notation in the tables, as well as rewritten parts of the paper to better stress the main goals and contributions.
>
> **Experiments only on ResNet Models**
> Indeed, all our experiments are conducted on ResNet models, which might prove a weakness. To accommodate this, we have included an additional experiment using VGG16 with CIFAR-100 in the revised manuscript. Unfortunately, due to time and computational constraints, we cannot reasonably include a Transformer model, as we also have other reviewers to accommodate. We might be able to conduct an experiment with the Transformer architecture by the camera-ready deadline on a dataset like TinyImageNet, but we are unable to scale up to larger datasets.
>
> **Is it a mistake that should be 'μ = 0.0, solid line'?**
> Thank you for mentioning this error, this was indeed a typo, which has since been resolved.
>
> Please let us know whether any other questions arise.
>
> Kind regards,
> the authors

---

### Review · Reviewer_15JU · 2025-03-01

**Summary Of Contributions:**

This paper investigates how different training settings—such as the influence of momentum and batch size—affect the instability of stochastic gradient descent (SGD). Prior research has linked SGD instability to the ability of a network to identify an optimal subnetwork, commonly referred to as the "winning ticket" in the lottery ticket hypothesis. Building on this foundation, the paper conducts an extensive numerical study to examine the transferability of these "winning tickets" across different datasets, particularly in scenarios with varying degrees of SGD instability. The findings provide insights into how different training approaches impact the generalizability of subnetworks and their effectiveness.

**Audience:**

Yes

**Claims And Evidence:**

Yes

**Requested Changes:**

See weakness.

**Strengths And Weaknesses:**

(**Strength.**) This paper provides an extensive study of the "winning" ticket found by IMP, which offers several promising and interesting directions: For example, (1) The paper conducts an extensive numerical study on the transferability of winning tickets to different datasets, showing that tickets extracted under more stable SGD conditions encode more generalizable features; (2) certain hyperparameter settings that improve generalization in dense networks can increase SGD instability, making it harder to extract winning tickets; (3) the paper also demonstrates that it is possible to find high-performing sparse subnetworks without pertaining.

(**Weakness.**)  Here are some major concerns of this paper:

(1) I would like to see that authors can provide some theoretical analysis of how different hyperparameter selections can affect instability. Also, I believe there are theoretical frameworks for LTH, which are largely missing in this paper. It is also better to talk about these theoretical frameworks and build connections with this manuscript.

(2)  The model architecture is limited to ResNet. Given that the paper references LLMs, it would be valuable to verify whether similar insights hold for Transformer-based models

(3) I believe the presentation of this paper needs improvement. For example, I do not fully understand how the conclusions in Section 6.1 are drawn. It would be helpful to include references to previous numerical results to clarify and support these conclusions.

---

> ### Author Response · Authors · 2025-03-04
>
> Dear reviewer 15JU,
>
> We thank you for your review. Please find below our responses.
>
> **Limited model architectures.**
> We have already added an additional experiment with VGG-16 on CIFAR-100 to accommodate reviewer gBsZ in the new version of the manuscript. While transformer models can certainly lead to additional insight in the applicability of our findings, unfortunately due to time constraints and limited computational resources, we cannot generate these results within the rebuttal period. We can strive to add transformer results for the camera-ready on a dataset such as TinyImageNet, but we cannot upscale to larger datasets due to the expensive nature of the experiments.
>
> **Improved presentation.**
> We have rewritten parts of the manuscript to improve the presentation of the results.
>
> **Theoretical analysis.**
> Indeed, most of our paper is currently focused on empirical results. To the best of our knowledge there has not yet been any analysis on how SGD instability is affected by different hyperparameters, as the field of linear mode connectivity is quite small. As for theoretical frameworks for the Lottery Ticket Hypothesis, we are aware of some theoretical works that show the maximum amount of pruning that can be done for a network to retain the winning property such as in [Paul22], and in [Zhang23]. When time allows us, we could set up an experiment, to test the different maximal pruning ratios for different tickets, such as in section 3.3. of [Paul22], however we do not yet have a notion of the computational requirements to run those in our settings. Additionally, several theoretical frameworks [Burkholz22a, Burkholz22b, daCunha22] exist for the Strong Lottery Ticket Hypothesis, however that setting is different from ours.
>
> Don't hesitate to reach out if any other questions arise.
>
> Kind regards,
>
> the authors
>
> References:
>
> [Paul22] : Paul et al. 2022 “Unmasking the Lottery Ticket Hypothesis: What's Encoded in a Winning Ticket's Mask?”
>
> [Zhang23] : Zhang et al. 2023 “How Sparse Can We Prune A Deep Network: A Geometric Viewpoint”
>
> [Burkholz22a] : Burkholz 2022 “Convolutional and Residual Networks Provably Contain Lottery Tickets”
>
> [Burkholz22b] : Burkholz 2022 “Most Activation Functions Can Win the Lottery Without Excessive Depth”
>
> [daCunha22]: daCunha et al. 2022 “Proving the strong lottery ticket hypothesis for convolutional neural networks”

---

### Decision · Action_Editor_HrPa · 2025-04-04

**Recommendation:** Reject

**Comment:**

In their official recommendations overall the reviewer's recommendations were negative: two of the four reviewers recommended "Learning Reject", one reviewer "Reject", and one "Leaning Accept". The lone positive recommendation only pointed to novelty of the research direction in this recommendation, but importantly did not champion the paper in the identified weaknesses all the other reviewers clearly agreed on, but also mirrored these concerns. Of the reviewers that recommended rejection, the common reasoning was a lack of clear and conclusive evidence for the claims made in the paper, and the poor presentation of the paper. While the reviewers were grateful to the authors for their rebuttal feedback, the reviewers did not believe these to address their main concerns about the lack of conclusive evidence for the strong claims made in the work. Particular items of concern included the limited model architectures and datasets (specifically the lack of larger models and large datasets where the main instability challenge for LTH lies), or a lack of theoretical explanations.

Concerns from the reviewers that did not impact the AE assessment of claims and evidence here include the concerns on novelty and lack of theoretical justification alone. The novelty concerns of Reviewer 2Zz7 lacked a clear citation or reference of work demonstrating lack of novelty, and novelty is not a decision criteria of TMLR itself. The lack of a theoretical justification *alone* would not be a concern if the work presented conclusive empirical evidence.

In line with reviewer recommendations, and in the light of the significant changes required to address the reviewer's concerns, it is appropriate to reject this submission at this time. Given the consensus around the novelty and potential of the research direction, and some interesting preliminary findings, the AE has decided to allow resubmission at a later time with a major revision. However, it should be emphasized that such a submission would need to be substantially revised to clearly address the reviewer's concerns.

**Audience:**

Most of the reviewers with the exception of Reviewer 2Zz7, agreed on the importance of the problem the authors are motivated to address, and the novelty and potential of the research direction proposed, for example Reviewer 15JU:
>  This paper provides an extensive study of the "winning" ticket found by IMP, which offers several promising and interesting directions

and Reviewer gBsZ:

> The paper tries to study an important problem in sparse training/LTH --- how SGD noise effect the LTH.

The AE agrees that the topic is in the scope for readers of TMLR, as the instability/requirement for pretraining of the Lottery Ticket Hypothesis (LTH) is a topic of active research within the community.

However, most reviewers did also agree that the paper itself could be unclear and poorly presented as-is, and requires significant revision. As Reviewer 2Zz7 states:

> The main claim of the paper is unclear. ... It is unclear which parts of the paper present novel contributions. The Introduction consists entirely of a review of existing studies, making it difficult to understand the motivation, research direction and new contributions of this research.

and Reviewer 15JU:

> I believe the presentation of this paper needs improvement. ...

The lack of clarity in the presentation of the work unfortunately obscures the claims of the work, and significantly reduces the ability of the work to reach its audience. This also came up in the reviewer's recommendations as a significant factor in their decisions.

In conclusion, while there is potential for the work to find an audience in TMLR, the current state of the submission, along with the lack of clarity in the results and claims themselves, means the audience rating is evaluated as borderline.

**Claims And Evidence:**

Most reviewers leaned to rejection after rebuttal with the majority of concerns resting on the consensus of all reviewers of a lack of conclusive and sufficient evidence for the claims being made, and in some cases the lack of clarity over those claims themselves. Common concerns related by the reviewers on the negative side included:

* The lack of large scale models or datasets, despite this being the setting in which instability for LTH training was originally identified.
* Insufficient results to conclusively make the claims being made.
* Lack of theoretical justification (in the absence of sufficient empirical evidence).
* Lack of different model architectures (outside of CNNs)
* The presentation of the paper itself.

While the authors did actively engage in the rebuttal, including providing results outside of ResNets alone on the architecture front, and improving the clarify of some of the experimental details and results, overall the reviewers did not find this enough to sufficiently address the above concerns.

On the positive side, one reviewer highlighted the novelty of the research direction as a reason for leaning towards acceptance, and another identified the related work and background to be generally well presented.

**Resubmission Of Major Revision:**

The authors may consider submitting a major revision at a later time.